# Why Attention Patterns Exist: A Unifying Temporal Perspective Analysis

**Qingyue Yang**[1]*, **Jie Wang**[1]†, **Xing Li**[2]‡, **Yinqi Bai**[1], **Xialiang Tong**[2], **Huiling Zhen**[2],
**Jianye Hao**[3], **Mingxuan Yuan**[2], **Bin Li**[1]

[1]MoE Key Laboratory of Brain-inspired Intelligent Perception and Cognition,
University of Science and Technology of China
[2]Huawei Technologies Co., Ltd.
[3]College of Intelligence and Computing, Tianjin University
`yangqingyue@mail.ustc.edu.cn`, `li.xing2@huawei.com`

## Abstract

Attention patterns play a crucial role in both training and inference of large language models (LLMs). Prior works have identified individual patterns such as retrieval heads, sink heads, and diagonal traces, yet these observations remain fragmented and lack a unifying explanation. To bridge this gap, we introduce **Temporal Attention Pattern Predictability Analysis (TAPPA), a unifying framework that explains diverse attention patterns by analyzing their underlying mathematical formulations** from a temporally continuous perspective. TAPPA both deepens the understanding of attention behavior and guides inference acceleration approaches. Specifically, TAPPA characterizes attention patterns as predictable patterns with clear regularities and unpredictable patterns that appear effectively random. Our analysis further reveals that this distinction can be explained by the degree of query self-similarity along the temporal dimension. Focusing on the predictable patterns, we further provide a detailed mathematical analysis of three representative cases through the joint effect of queries, keys, and Rotary Positional Embeddings (RoPE). We validate TAPPA by applying its insights to KV cache compression and LLM pruning tasks. Across these tasks, a simple metric motivated by TAPPA consistently improves performance over baseline methods. The code is available at `https://github.com/MIRALab-USTC/LLM-TAPPA`.

## 1 Introduction

Attention patterns matter for both LLM training and inference (Jiang et al., 2024; Geng et al., 2025; Li et al., 2025; Yang et al., 2025). Prior studies have shown that attention heads exhibit structured and reusable forms, such as streaming heads, retrieval heads, and sink heads (Xiao et al., 2023; 2024). Understanding why such patterns emerge is critical for a deeper conceptual understanding of the attention mechanism and can directly inform the design of architectures and inference strategies that improve efficiency and robustness, for example, cache compression and pruning.

A substantial body of recent research has empirically analyzed attention behavior. Prior analyses typically focus on a single phenomenon, for example, the attention sink at the first token (Gu et al., 2024) or diagonal traces linked to high-frequency components of RoPE (Barbero et al., 2025). Other studies categorize heads by functional roles, such as retrieval and streaming (Xiao et al., 2023; 2024). Despite these advances, it remains unclear what factors determine which attention pattern a head will adopt under the same attention formulation. Our goal is to uncover a unifying underlying mechanism that explains the emergence of these diverse patterns.

---

*This work was done when Qingyue Yang was an intern at Huawei.
‡Project lead.
†Corresponding author. Email: jiewangx@ustc.edu.cn.

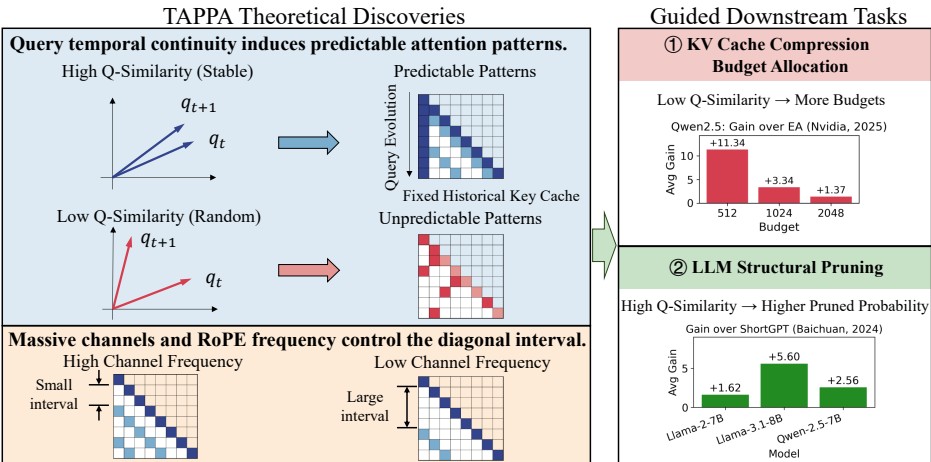

Figure 1: Overview of Temporal Attention Pattern Predictability Analysis (TAPPA) Framework. Left: Theoretical discoveries. Query self-similarity (q-similarity) affects the predictable and unpredictable patterns. Within the periodic sequential pattern, the slash interval is affected by the joint effect of queries, keys and RoPE. Right: Q-similarity is applied to downstream tasks and achieves consistent improvements.

To address this gap, we adapt a temporal view of auto-regressive inference and analyze how attention evolves over time. During inference, a transformer LLM generates each token from the previously generated sequence, so the hidden states and attention scores across steps can be regarded as a temporal series. We then isolate the source of temporal variation in attention along the time axis. Conditioned on the past keys being fixed, changes in the attention distribution across steps are determined by the evolution of the query vector. In this interaction, a few embedding channels may dominate the inner product (Sun et al., 2024; Liu et al., 2024), which determines the shape of the attention pattern. Figure 1 provides an illustration of how changes in queries and dominant embedding channels reshape the attention pattern.

Guided by the temporal view, we propose **Temporal Attention Pattern Predictability Analysis (TAPPA)**, a unified framework that interprets attention patterns through the temporal behavior of queries and the response of the RoPE channels. We view the sequence of query vectors and the associated attention distributions as a time series and characterize them using the notion of continuity. We mathematically show that temporal continuity of queries, measured by their self-similarity, is the key factor distinguishing *predictable patterns*, characterized by clear regularities, and *unpredictable patterns* that lack stable regularities. Within the predictable regime, we further provide theoretical conditions for three representative patterns with the joint effect of queries, keys, and RoPE. **Re-access patterns**, where an attention head repeatedly focuses on a small set of tokens, require high query self-similarity and a favorable initial query-key geometry. **Sequential patterns**, which appear as diagonals, are driven by high self-similarity in both queries and keys. In this case, we provide a sufficient condition that does not require attributing the phenomenon exclusively to high-frequency components (Barbero et al., 2025). **Seasonal patterns**, which periodically repeat focus, arise when input periodicity combines with the periodic nature of dominant embedding channels. Since the computing attention from queries, keys, and RoPE is a common design in transformer-based models, TAPPA both unifies diverse attention patterns and is broadly applicable across LLMs.

To validate TAPPA, we evaluate it on downstream tasks. Prior works have shown that attention patterns are closely linked to a model's representational capacity (Li et al., 2025; Xiao et al., 2024) and can guide compression. Stable temporal behavior indicates redundant or predictable attention allocation, which can be exploited for selective retention or pruning. Building on this view, we focus on two complementary compression settings: KV cache compression for stored states and LLM pruning for model weights. In both cases, a simple metric q-similarity consistently outperforms baselines, demonstrating that these principles are practically useful.

In summary, our contributions are as follows: (1) We introduce TAPPA, which provides the first systematic analysis of the shapes of attention patterns from a unifying temporal perspective, analyz-

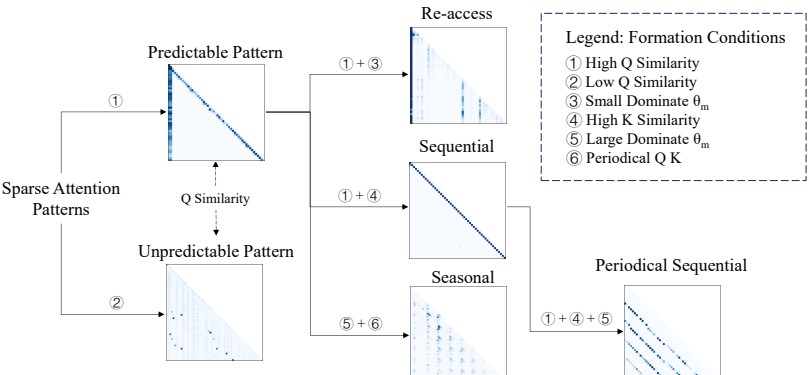

Figure 2: TAPPA explains the formation of sparse attention patterns from a temporal continuity perspective. We first establish the fundamental Predictable and Unpredictable patterns in Sec. 4. We then detail the conditions that form the Re-access (Sec. 5.1), Sequential (Sec. 5.2), Seasonal (Sec. 5.4), and Periodical Sequential (Sec. 5.3) patterns in their dedicated sections.

ing unpredictable patterns alongside three predictable types: re-access, sequential, and seasonal. (2) Theoretically, we demonstrate that stable patterns emerge from the continuity of queries and keys combined with the RoPE mechanism. (3) We identify periodic sequential diagonals and explain them as a consequence of the RoPE rotation period of the dominant channel. (4) We apply insights derived from TAPPA to downstream tasks, including KV cache compression and LLM pruning, achieving accuracy improvements.

## 2 RELATED WORK

### 2.1 ATTENTION PATTERNS

The sparse nature of attention mechanisms in Large Language Models (LLMs) is well-documented, giving rise to distinct, recurring patterns. Prior work has largely focused on identifying these patterns and using them for inference optimization. For instance, one widely discussed pattern is the *attention sink*, where high attention scores are consistently assigned to the initial tokens (Xiao et al., 2023), attracting significant research interest and analysis from various perspectives (Gu et al., 2024; Yu et al., 2024; Cancedda, 2024). Xiao et al. (2023) also highlighted the importance of attention to *recent tokens*, which form a distinct diagonal trace in the attention map. The structured nature of these patterns has been widely exploited for KV cache compression and inference optimization by various methods, such as Minference (Jiang et al., 2024), H2O (Zhang et al., 2024), SnapKV (Li et al., 2024), DuoAttention (Xiao et al., 2024), and KVTuner (Li et al., 2025). Alongside these structured patterns, other works have identified *retrieval heads* (Wu et al., 2024; Xiao et al., 2024). These heads appear to scan the entire context for semantically relevant information, resulting in seemingly random attention maps that are crucial for long-context reasoning and factuality (Xiao et al., 2024). However, these observations have remained largely fragmented, lacking a unifying theory to explain the co-existence and emergence of these diverse patterns.

### 2.2 THE ROLE OF POSITIONAL ENCODING

A growing body of work has sought a mechanistic explanation for these patterns by examining the role of Rotary Positional Embeddings (RoPE) (Su et al., 2024). Research has shown a direct link between RoPE's frequency components and specific pattern shapes. For instance, high-frequency components in RoPE have been demonstrated to be responsible for the formation of diagonal or previous-token patterns (Barbero et al., 2025). Conversely, other studies suggest that low-frequency components, or specific *outlier channels* with large magnitudes, may contribute to the emergence of attention sinks by creating a rotational offset that favors certain positions (Jonasson, 2025). While these studies provide crucial insights into how positional encoding shapes attention, they often analyze RoPE's effects in isolation, without fully modeling its interaction with the dynamic content of the query and key vectors.

## 2.3 THE INFLUENCE OF INPUT DYNAMICS

A parallel line of research investigates how the properties of the input tokens themselves influence attention patterns. AttentionPredictor (Yang et al., 2025) proposed that the temporal continuity of queries is a key driver for pattern formation, though it did not provide a deep mathematical analysis or consider the interplay with RoPE. Other works have corroborated the importance of input features, suggesting that attention sinks may arise from specific query-key angular relationships that are independent of position (Gu et al., 2024). Similarly, the continuity of queries between layers and constant massive channels of keys has also been noted (Lee et al., 2024; Liu et al., 2024), hinting at the inherent temporal consistency within the model. However, this line of inquiry has yet to be formally connected with the rotational effects of RoPE to provide a complete picture.

In this work, we bridge the gap between these latter two perspectives. We propose TAPPA, a unifying theoretical framework that explains how input dynamics and positional encoding together influence attention patterns. Specifically, we demonstrate that variations in query self-similarity over time, when coupled with the rotational mechanics of RoPE, can mathematically account for the diverse patterns observed in prior works.

## 3 BACKGROUND

**Attention Mechanism.** At the decoding step $t$, let the query be $q_t \in \mathbb{R}^d$, the key matrix $K = [k_1, \ldots, k_T]^\top \in \mathbb{R}^{T \times d}$ with $k_j \in \mathbb{R}^d$, and the unnormalized logits

$$a_{t,j} = q_t^\top R_{t-j} k_j, \quad a_t \in \mathbb{R}^T, \tag{1}$$

where $R_{t-j}$ is the *Rotary Positional Embedding* (RoPE) operator that rotates vector $k_j$ by a relative phase proportional to $(t - j)$.

The attention distribution is then

$$A_t = \text{softmax}(a_t). \tag{2}$$

Since the softmax function is monotonic with respect to the logits, it preserves their relative order across positions. Therefore, for clarity, our discussion focuses on the logits $a_t$, and the resulting conclusions directly extend to the final attention distribution $A_t$.

**RoPE.** RoPE encodes relative position information by applying channel-wise 2D rotations to pairs of embedding dimensions. For feature pair $(m, m + d//2)$ at position $m$, the rotation is

$$R_{n,m} = \begin{pmatrix} \cos(n\theta_m) & -\sin(n\theta_m) \\ \sin(n\theta_m) & \cos(n\theta_m) \end{pmatrix}, \tag{3}$$

where $\theta_m = c^{-2m/d}$ is the frequency of the $m$-th channel, $d$ is the hidden dimension, and $c$ is a hyperparameter. While the original RoPE paper (Su et al., 2024) proposed pairing adjacent dimensions, this half-split pairing scheme is adopted by large-scale models like Llama and Qwen2 for greater computational efficiency.

Thus, for a query $q_t$ and key $k_i$, the RoPE-augmented attention score on channel $m$ is

$$a_{t,i}^{(m)} = q_t^{(m)\top} R_{t-i,m} k_i^{(m)}, \tag{4}$$

where $q_t^{(m)} = (q_{t,2m}, q_{t,2m+1})^\top$.

**Decomposition View of Attention.** Using the RoPE formulation, the attention logits $a_{t,i}$ between $q_t$ and $k_i$ can be decomposed channel-wise. Let $q_t = \bigoplus_{m=1}^M q_t^{(m)}$ and $k_i = \bigoplus_{m=1}^M k_i^{(m)}$, where each pair $q_t^{(m)}, k_i^{(m)} \in \mathbb{R}^2$ corresponds to a frequency channel with angular frequency $\theta_m = c^{-2m/d}$. Then

$$a_{t,i} = \sum_{m=1}^M \|q_t^{(m)}\| \|k_i^{(m)}\| \cos\left(\phi_{t,i}^{(m)} + (i - t)\theta_m\right), \tag{5}$$

where $\phi_{t,i}^{(m)}$ denotes the angle between $q_t^{(m)}$ and $k_i^{(m)}$. This decomposition highlights how each frequency channel contributes additively to the overall attention score, and how temporal shifts $(i - t)$ are modulated by channel-dependent phases $\theta_m$.

**RoPE Key Property.** RoPE satisfies a relative-position identity:

$$R_m^\top R_n = R_{m-n}, \tag{6}$$

which ensures that attention depends only on the relative distance $(t - i)$, not absolute positions.

**Attention Patterns.** It is well-established that the attention mechanism is sparse and shows various patterns. In this work, we focus on these sparse attention patterns, especially in Llama-3.1-8B (Dubey et al., 2024) and Qwen-2.5-7B (Yang et al., 2024a) with GSM8K (Cobbe et al., 2021) and AIGC (SoftAge-AI, 2024) datasets.

# 4 WHY PREDICTABLE AND UNPREDICTABLE ATTENTION PATTERNS EXIST

Previous works mainly analyze and utilize attention patterns, including retrieval/streaming heads or A-shape/vertical-slash/block-sparse patterns, from the functionality or geometric morphology view. In contrast, TAPPA provides a new and unifying time-series analysis perspective to theoretically understand the existence of diverse attention patterns (Figure 2), utilizing the underlying attention mechanisms. Under TAPPA, attention patterns fall into two temporal categories: **predictable** and **unpredictable**. Predictable patterns exhibit temporal continuity across decoding steps or along the temporal dimension, where the indices of high attention scores evolve smoothly over time. Unpredictable patterns display irregular jumps with little temporal consistency. This distinction matters because temporal stability enables inference optimization: stable patterns can be anticipated and efficiently compressed in the KV cache, while unpredictable ones resist such treatment.

Empirically, retrieval attention heads exemplify the unpredictable case. Their attention often jumps across the entire context in a seemingly random fashion (Wu et al., 2024; Xiao et al., 2024; Li et al., 2025), which is crucial for retrieving semantically relevant information but undermines predictability. Predictable patterns, by contrast, correspond to heads that consistently attend to locally structured or repeatedly accessed tokens, reflecting stable model behaviors that are exploitable for compression and acceleration.

In TAPPA, the key differentiator behind these two regimes is **query self-similarity**, namely the similarity between queries at different time indices. *When successive queries remain close in representation space, attention indices change smoothly and produce predictable attention maps.* When queries drift strongly, the inequalities that define structured patterns are violated. Even with RoPE relative rotations, attention can jump unpredictably. To capture this distinction, we introduce a quantitative measure of query continuity, termed *q-similarity*.

In Appendix F.3, we further study the distribution of q-similarity across layers, heads, models, and datasets, and show that high-continuity heads are common but not universal. High q-similarity correlates with stable, predictable heads, while low q-similarity leads to retrieval-like, unpredictable behavior. Figure 3 shows the attention patterns of the two models with high and low q-similarity scores. It can be seen that patterns with high q-similarity are more stable, while patterns with low q-similarity are more random.

**Proposition 4.1.** *Let $q_t, q_{t+1} \in \mathbb{R}^d$ be consecutive queries, $K = [k_1, \ldots, k_T]^\top$ the key matrix, and define the logits*

$$a_{t,j} = q_t^\top R_{t-j} k_j, \qquad a_{t+1,j} = q_{t+1}^\top R_{t+1-j} k_j.$$

*If $q_{t+1} - q_t$ has a large norm and is not orthogonal to all rotated keys $\{R_{t+1-j}k_j\}$, then the difference between the logit vectors $a_t$ and $a_{t+1}$ is necessarily large. In particular, there exist constants $c_1, c_2 > 0$ such that*

$$\|a_{t+1} - a_t\|_\infty \geq c_1 \|q_{t+1} - q_t\| - c_2.$$

Proposition 4.1 demonstrates that while low q-similarity leads to more random patterns, high q-similarity is a necessary condition for predictable ones. In summary, q-similarity provides a quantitative indicator of whether an attention head behaves in a predictable or unpredictable manner. In the following sections, the theoretical analysis focuses on the predictable heads.



Figure 3: Attention patterns at high and low Query similarity on the Llama and Qwen models. Stable patterns emerge under high similarity, whereas low similarity results in random patterns. There are random bright dots of critical keys in the second and fourth figures.

## 5 PREDICTABLE ATTENTION PATTERNS

In this section, we provide a temporal perspective analysis on predictable attention patterns, which rely on the temporal continuity of queries. The **re-access** pattern occurs when queries are highly self-similar, with low-frequency RoPE components helping to maintain alignment with fixed keys. We also discuss how this analysis relates to the conditions described in prior work (Gu et al., 2024). **Sequential** patterns arise from the combination of high query and key similarity and the relative position property of RoPE. In some cases, **periodic sequential** patterns appear. We provide a clear calculation for the spacing between adjacent periods and verify it experimentally by varying the location of the dominant RoPE channel and the RoPE base parameter. Finally, we analyze a **seasonal** pattern with periodical queries and keys.

These predictable patterns are useful for LLM inference acceleration. Methods that exploit such temporal regularities, including Minference (Jiang et al., 2024), H2O (Zhang et al., 2024) and SnapKV (Li et al., 2024) can compress the KV cache with little loss in LLM performance, which empirically supports the claim that temporal stability is an important signal for effective KV compression (Jiang et al., 2024; Zhang et al., 2024; Li et al., 2024).

### 5.1 RE-ACCESS PATTERN

The re-access pattern describes repeated attention to a small set of key tokens, appearing as vertical lines in the attention map and often referred to as attention sink (Xiao et al., 2023). Prior work has attributed this phenomenon to query continuity (Yang et al., 2025) or to the small angle between the first key and all queries (Gu et al., 2024), while others observed its correlation with low-frequency RoPE rotations (Jonasson, 2025). However, these explanations are partial, and TAPPA provides a unified account of why they align under the same mechanism.

We propose that the stability of re-access pattern relies on two factors: (1) high self-similarity of consecutive queries, which prevents attention scores from drifting, and (2) the low-frequency components of RoPE, which preserve alignment between queries and fixed keys even as time $t$ increases.

**Theorem 5.1** (Vertical Stability of Attention). *Suppose the channel-wise decomposition (Background, Eq. 5) holds for the attention logits $a_{t,i}$. Assume that the queries evolve continuously in the sense that $\|q_{t+1} - q_t\| \leq \varepsilon$, while all keys $k_i$ remain fixed between steps $t$ and $t + 1$. Further assume the existence of a dominant low-frequency channel $m^\star$ whose weight $w_{m^\star}$ dominates the other channels, and whose RoPE frequency $\theta_{m^\star}$ is small. Then the per-key differences $a_{t+1,i} - a_{t,i}$ are uniformly small, and the attention logits are vertically stable.*

When queries vary little over time or decoding steps, the only source of temporal change in equation 5 is the RoPE-induced phase $(i - t)\theta_m$. If a dominant channel with small $\theta_m$ controls the sum, then shifting $t \mapsto t + 1$ changes the cosine term only marginally, hence $a_{t+1,i} \approx a_{t,i}$. This yields vertically aligned attention weights. The empirical validation of the dominant-channel assumption for the re-access head is in Appendix F.1.

**Connection to Attention Sink in the First Token.** A well-known empirical phenomenon is the *attention sink*, which typically appears at the first token position. Prior work Gu et al. (2024) observed that queries and keys at the initial position tend to have a very small angle, and attributed this alignment as the cause of the sink. TAPPA analysis provides a complementary explanation: from

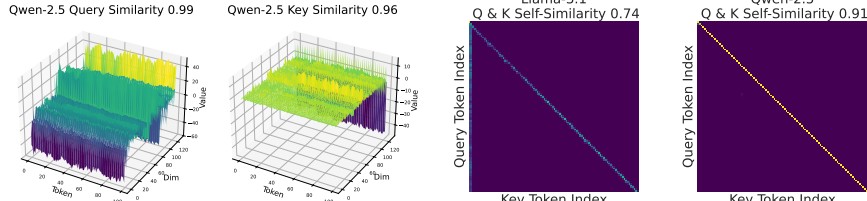

Figure 4: High self-similarity in Query (Q) and Key (K) matrices results in sequential attention patterns. An example from a Qwen-2.5 head (left) with high Q and K self-similarity (0.99 and 0.96) produces a strong diagonal pattern in the attention map (far right). This phenomenon is also observed in Llama-3.1 (center right).

the decomposition in Equation 5, when the angle $\phi_{t,i}^{(m)}$ between $q_t^{(m)}$ and $k_i^{(m)}$ is small, the cosine term $\cos(\phi_{t,i}^{(m)} + (i - t)\theta_m)$ is close to 1. Consequently, the logit contribution from that channel approaches its maximum possible value $\|q_t^{(m)}\|\|k_i^{(m)}\|$, making the overall attention score $a_{t,i}$ large. This alignment effect explains why high attention scores often emerge at positions where $q$ and $k$ are nearly aligned, particularly at the first token.

## 5.2 SEQUENTIAL PATTERN

Sequential patterns exhibit a shifting focus across tokens, typically progressing step by step along the sequence. The diagonal slash often observed near the main diagonal is commonly attributed to positional heads, which attend to tokens at fixed relative offsets. We argue that the sequential pattern arises from the combined effect of both high q-similarity and k-similarity and the relative-position property of RoPE.

**Theorem 5.2** (Sequential Patterns under High Self-similarity). *Under the RoPE relative-position encoding, suppose queries and keys both exhibit high self-similarity, in the sense that*

$$\|q_{t+1} - q_t\| \le \varepsilon, \qquad \|k_{i+1} - k_i\| \le \varepsilon$$

*for sufficiently small $\varepsilon > 0$. Then the attention logits satisfy*

$$|a_{t+1,i+1} - a_{t,i}| \le C\varepsilon,$$

*for some constant $C > 0$. Consequently, the attention logits exhibit approximate shift-invariance along the $(+1, +1)$ diagonal, giving rise to sequential patterns in the attention map.*

RoPE encodes relative positions through rotations. When queries and keys vary little across steps, this rotation structure preserves their interactions under a simultaneous shift. As a result, attention scores propagate along the $(+1, +1)$ diagonal, producing sequential (slash-like) patterns.

**Empirical Results.** High self-similarity in both query and key representations is a sufficient condition for the emergence of Sequential patterns. Figure 4 illustrates the patterns of heads with high query similarity and high key similarity, all of which clearly exhibit diagonal structures. Empirical results for separating the roles of input dynamics and RoPE are in Appendix F.2.

## 5.3 PERIODICITY OF SEQUENTIAL PATTERNS

Empirically, we sometimes observe multiple parallel diagonal lines in attention maps, with a roughly constant spacing between adjacent lines (*periodic sequential pattern*). We attribute this periodicity to the rotation angle of the dominant RoPE channel.

**Theorem 5.3** (Periodic Sequential Pattern from a Dominant RoPE Channel). *If a sequential pattern arises and the corresponding key exhibits a massive channel at index $m^\star$, then the spacing between adjacent diagonals is determined by the rotation frequency of that channel:*

$$T = \frac{2\pi}{\theta_{m^\star}} = 2\pi c^{2m^\star/d}. \tag{7}$$

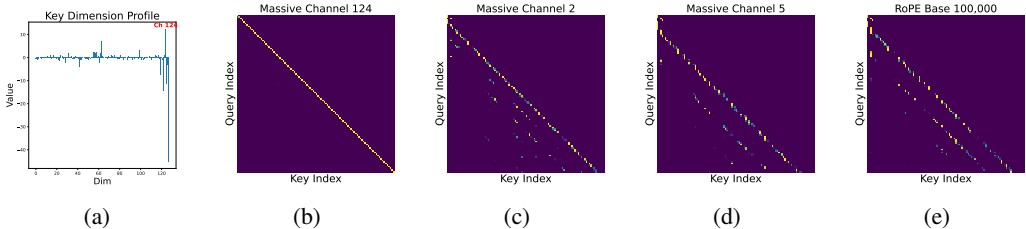

Figure 5: An illustration of how RoPE configuration affects attention patterns. (a) and (b) show a sequential pattern with a dominant channel at $m = 124$. In (c) and (d), we manually change the dominant channel to higher frequencies ($m = 2$ and $m = 5$), which causes periodic diagonals to emerge. In (e), we change the RoPE base from $c = 1,000,000$ to $c = 100,000$ with $m = 5$.

**Intuition.** When the massive channel is $m^\star$, the attention score is dominated by that component:

$$a_{t,j} \approx \|q_t^{(m^\star)}\| \|k_j^{(m^\star)}\| \cos\left(\phi_{t,j}^{(m^\star)} + (j - t)\theta_{m^\star}\right).$$

This term is a cosine function of the relative offset $(j-t)$ with angular frequency $\theta_{m^\star}$. Consequently, the diagonal lines in the attention map exhibit a regular repetition with period $T = 2\pi/\theta_{m^\star}$, as given in equation 7. Since $\theta_{m^\star} = c^{-2m^\star/d}$, higher channel indices $m^\star$ correspond to lower angular frequencies and therefore to greater spacing between adjacent diagonals.

We validate the theoretical mechanism with controlled manipulations on learned key vectors. We consider two axes of intervention: (i) relocating the massive channel across different indices, and (ii) varying the RoPE base hyperparameter $c$.

**Relocating the massive channel.** We first analyze a key vector $k_j$ whose attention map exhibits a single diagonal, as shown in Figure 5 (b). We identify its massive channel at index $m^\star = 124$ as shown in Figure 5 (a). Given the Qwen2.5 RoPE hyperparameters (base $c = 1,000,000$, dimension $d = 128$), this high-index channel corresponds to an extremely low angular frequency. Its theoretical period is $T = 2\pi c^{2m^\star/d} \approx 2.4 \times 10^6$, a value so large that no repetition can be observed within a practical context window.

To demonstrate the relationship between channel frequency and periodicity, we experimentally relocate this massive channel to different target indices $m$, recomputing the RoPE-augmented attention for each case. The resulting attention maps with $m = 2$ and $m = 3$, visualized in Figure 5 (c) and (d), show that periodic diagonals emerge as the massive channel is moved to lower-index, higher-frequency positions. Specifically, as the channel index $m$ decreases, the angular frequency $\theta_m$ increases, shortening the period $T$ and making the diagonals denser. This confirms the first finding of TAPPA: **observable periodic diagonals require the key's massive channel to reside in a high-frequency (low-index) position.**

Furthermore, we observe that even for high-frequency channels, the diagonal patterns fade over long distances. This occurs because the self-similarity between queries and keys naturally diminishes as their relative distance increases, which disrupts the continuity required to sustain the pattern.

**Varying the RoPE base** $c$**.** Independent of the channel index, the choice of RoPE base also controls the periodicity. To isolate this effect, we keep the same dominant channel $m^\star = 5$ and repeat the above procedure for different values of the base (e.g., $c = 1,000,000$ and 100,000 in Figure 5 (d) and (e)). Since the channel frequency is given by $\theta_m = c^{-2m/d}$, decreasing $c$ directly increases $\theta_m$ and hence reduces the diagonal period $T = 2\pi/\theta_m$.

## 5.4 SEASONAL PATTERN

Seasonal patterns arise when attention maps repeat with a fixed periodicity. This periodicity can manifest along either the temporal axis or the spatial axis. Due to the periodicity of the hidden states, the periodicity of queries and keys is often aligned, so temporal and spatial repetitions typically occur simultaneously and share the same period. We argue that the underlying cause of the seasonal pattern is that queries and keys exhibit periodicity, which is preserved and sometimes amplified by

Table 1: The evaluation results on the LongBench dataset across different KV cache budgets.

| Budget | Method | Single-DocumentQA | | | Multi-DocumentQA | | | Summary | | | Few-shot Learning | | | Synthetic | | Code | | Average↑ |
|---|---|---|---|---|---|---|---|---|---|---|---|---|---|---|---|---|---|---|
| | | NrtvQA | Qasper | MF-en | HotpotQA | 2WikiMQA | Musique | GovReport | QMSum | MultiNews | TREC | TriviaQA | SAMSum | PCount | PRe | Lcc | RB-P | |
| | | | | | | | | Llama-3.1-8B | | | | | | | | | | |
| Full | Full | 31.06 | 45.43 | 53.78 | 55.04 | 47.14 | 31.29 | 34.87 | 25.33 | 27.49 | 72.50 | 91.25 | 43.81 | 6.00 | 99.50 | 63.36 | 56.65 | 49.06 |
| 512 | StreamingLLM | 25.64 | 27.48 | 33.30 | 47.36 | 40.06 | 24.80 | 23.16 | 20.80 | 22.85 | 57.50 | 87.60 | 42.08 | 6.50 | 97.00 | 60.51 | 51.28 | 41.75 |
| | H2O | 27.76 | 29.01 | 44.75 | 52.78 | 44.31 | 29.22 | 24.71 | 23.11 | 24.56 | 54.50 | 91.38 | 42.10 | 6.36 | 99.00 | 62.30 | 54.33 | 44.39 |
| | SnapKV | 30.76 | 42.03 | 52.13 | 54.15 | 46.14 | 30.51 | 24.98 | 24.24 | 24.65 | 64.00 | 92.05 | 42.04 | 6.08 | 99.50 | 62.62 | 54.90 | 46.92 |
| | PyramidKV | 30.47 | 42.15 | 52.17 | 54.67 | 45.25 | 30.60 | 25.00 | 24.33 | 24.51 | 62.50 | 91.24 | 41.67 | 5.95 | 99.50 | 61.58 | 53.89 | 46.59 |
| | CAKE | 31.82 | 42.99 | 51.65 | 54.37 | 46.89 | 30.73 | 26.36 | 24.94 | 25.27 | 63.50 | 91.54 | 42.52 | 6.33 | 99.50 | 62.30 | 54.30 | 47.19 |
| | TAPPA | 29.47 | 42.66 | 51.63 | 54.53 | 46.64 | 30.81 | 25.48 | 24.57 | 24.71 | 62.50 | 92.35 | 42.42 | 6.25 | 99.50 | 64.56 | 57.35 | 47.21 |
| 1024 | StreamingLLM | 26.64 | 30.77 | 35.59 | 47.31 | 42.03 | 24.17 | 25.81 | 21.31 | 25.66 | 63.50 | 88.84 | 42.76 | 6.50 | 88.00 | 61.31 | 53.47 | 42.73 |
| | H2O | 29.57 | 36.15 | 45.94 | 54.43 | 44.81 | 29.04 | 27.64 | 23.31 | 26.47 | 62.00 | 91.43 | 43.14 | 6.36 | 99.00 | 62.24 | 55.74 | 46.11 |
| | SnapKV | 30.95 | 44.74 | 52.58 | 55.09 | 46.83 | 30.37 | 27.87 | 24.57 | 25.99 | 68.50 | 92.03 | 42.60 | 6.50 | 99.50 | 63.00 | 56.50 | 47.95 |
| | PyramidKV | 30.54 | 43.64 | 52.73 | 55.29 | 46.29 | 31.28 | 27.53 | 24.50 | 26.00 | 68.00 | 92.09 | 41.75 | 6.05 | 99.50 | 62.35 | 55.44 | 47.69 |
| | CAKE | 30.88 | 44.95 | 52.38 | 55.49 | 46.99 | 30.82 | 28.68 | 24.91 | 26.39 | 69.00 | 91.94 | 42.60 | 6.00 | 99.50 | 62.65 | 56.89 | 48.13 |
| | TAPPA | 30.77 | 44.94 | 52.14 | 55.43 | 46.99 | 31.16 | 28.72 | 24.90 | 26.65 | 69.50 | 91.95 | 42.38 | 6.00 | 99.50 | 64.99 | 58.84 | 48.43 |
| 2048 | StreamingLLM | 27.40 | 36.91 | 37.85 | 49.23 | 44.66 | 24.31 | 28.57 | 21.67 | 27.12 | 67.50 | 90.98 | 42.49 | 6.12 | 87.00 | 63.06 | 55.32 | 44.39 |
| | H2O | 29.65 | 39.53 | 48.64 | 54.23 | 46.50 | 29.28 | 29.97 | 23.68 | 27.21 | 68.50 | 91.48 | 43.06 | 6.11 | 99.50 | 63.06 | 56.91 | 47.30 |
| | SnapKV | 30.99 | 45.06 | 53.15 | 55.25 | 46.56 | 30.78 | 30.24 | 24.63 | 27.32 | 70.50 | 91.48 | 42.37 | 6.00 | 99.50 | 63.28 | 56.86 | 48.32 |
| | PyramidKV | 31.13 | 45.06 | 53.80 | 55.78 | 46.59 | 30.89 | 30.25 | 24.82 | 27.35 | 71.00 | 91.65 | 42.62 | 6.00 | 99.50 | 63.27 | 56.44 | 48.51 |
| | CAKE | 30.79 | 45.83 | 53.57 | 55.50 | 46.60 | 30.47 | 31.12 | 24.67 | 27.16 | 70.50 | 91.48 | 43.48 | 6.00 | 99.50 | 64.26 | 56.64 | 48.43 |
| | TAPPA | 30.70 | 45.69 | 53.06 | 55.49 | 46.68 | 30.94 | 30.54 | 24.65 | 27.12 | 71.00 | 91.65 | 43.00 | 6.00 | 99.50 | 64.93 | 58.80 | 48.73 |
| | | | | | | | | Qwen2.5-7B | | | | | | | | | | |
| Full | Full | 29.05 | 43.34 | 52.52 | 57.59 | 47.05 | 30.24 | 31.78 | 23.64 | 23.96 | 72.50 | 89.47 | 45.61 | 8.50 | 100.00 | 59.61 | 67.12 | 48.87 |
| 512 | StreamingLLM | 19.82 | 25.40 | 35.57 | 43.24 | 39.18 | 18.59 | 25.45 | 19.07 | 22.33 | 58.50 | 71.13 | 32.29 | 8.00 | 23.00 | 46.18 | 49.01 | 33.55 |
| | H2O | 26.83 | 34.17 | 41.43 | 50.80 | 41.83 | 22.82 | 25.57 | 21.35 | 22.03 | 60.50 | 84.67 | 45.86 | 8.00 | 95.50 | 59.11 | 64.66 | 44.07 |
| | SnapKV | 28.94 | 40.70 | 50.40 | 55.80 | 44.21 | 27.83 | 24.42 | 22.74 | 21.07 | 66.50 | 86.56 | 44.14 | 8.00 | 99.50 | 59.17 | 64.22 | 45.51 |
| | PyramidKV | 27.33 | 38.04 | 50.38 | 55.73 | 44.28 | 27.12 | 22.24 | 21.86 | 19.54 | 66.00 | 86.36 | 43.69 | 8.00 | 99.00 | 57.59 | 62.09 | 45.58 |
| | CAKE | 28.97 | 39.46 | 50.40 | 54.80 | 44.70 | 28.02 | 23.90 | 22.35 | 20.74 | 55.00 | 86.91 | 44.92 | 8.00 | 99.50 | 57.06 | 64.26 | 45.56 |
| | TAPPA | 28.97 | 39.40 | 50.46 | 55.48 | 44.47 | 28.02 | 23.99 | 22.87 | 20.72 | 55.00 | 87.02 | 44.66 | 8.00 | 100.00 | 59.04 | 64.39 | 45.78 |
| 1024 | StreamingLLM | 22.72 | 29.42 | 31.47 | 43.57 | 38.18 | 17.99 | 24.33 | 19.47 | 22.46 | 61.00 | 87.53 | 43.79 | 8.50 | 34.00 | 55.17 | 58.43 | 37.38 |
| | H2O | 26.45 | 34.94 | 40.49 | 48.63 | 42.02 | 22.27 | 25.67 | 20.90 | 22.41 | 59.00 | 87.83 | 45.07 | 8.50 | 98.48 | 59.77 | 63.88 | 44.14 |
| | SnapKV | 29.24 | 41.61 | 50.93 | 57.60 | 45.50 | 29.39 | 25.63 | 22.06 | 22.26 | 65.50 | 88.92 | 44.65 | 8.50 | 100.00 | 58.16 | 65.30 | 47.27 |
| | PyramidKV | 29.34 | 38.60 | 50.17 | 55.67 | 45.12 | 27.82 | 23.26 | 22.16 | 20.55 | 62.50 | 86.85 | 43.26 | 8.50 | 100.00 | 57.76 | 61.99 | 45.85 |
| | CAKE | 29.47 | 42.71 | 52.12 | 56.11 | 46.41 | 29.13 | 26.86 | 23.12 | 22.72 | 67.50 | 89.23 | 45.24 | 8.50 | 100.00 | 59.11 | 64.24 | 47.70 |
| | TAPPA | 29.64 | 43.59 | 51.53 | 56.90 | 45.86 | 29.43 | 26.64 | 22.57 | 23.00 | 65.50 | 89.48 | 45.24 | 8.00 | 100.00 | 60.04 | 66.06 | 47.72 |
| 2048 | StreamingLLM | 23.18 | 36.93 | 45.64 | 45.30 | 40.10 | 19.74 | 28.49 | 20.57 | 23.50 | 68.00 | 74.41 | 33.06 | 8.00 | 18.00 | 54.50 | 53.73 | 37.07 |
| | H2O | 28.55 | 40.20 | 47.45 | 53.49 | 44.44 | 27.00 | 28.93 | 22.66 | 23.79 | 63.50 | 88.50 | 46.08 | 8.00 | 100.00 | 61.06 | 67.50 | 46.95 |
| | SnapKV | 29.11 | 41.53 | 52.05 | 57.17 | 46.26 | 30.69 | 29.49 | 23.23 | 23.64 | 71.50 | 89.17 | 45.49 | 8.00 | 100.00 | 60.92 | 67.94 | 48.12 |
| | PyramidKV | 28.39 | 43.36 | 51.83 | 56.75 | 45.60 | 30.50 | 26.90 | 23.08 | 23.38 | 71.00 | 88.22 | 45.01 | 8.00 | 100.00 | 61.36 | 67.37 | 48.17 |
| | CAKE | 29.08 | 43.35 | 51.92 | 57.20 | 45.77 | 30.26 | 29.35 | 23.46 | 23.59 | 69.00 | 89.37 | 45.37 | 8.00 | 100.00 | 59.35 | 67.88 | 48.31 |
| | TAPPA | 29.18 | 44.03 | 52.24 | 57.36 | 45.77 | 30.19 | 29.14 | 23.31 | 23.62 | 69.00 | 89.47 | 44.99 | 8.00 | 100.00 | 60.95 | 68.06 | 48.46 |

RoPE through its relative-position encoding. Although the query condition does not exhibit temporal continuity, the pattern remains predictable over time and is therefore a predictable pattern.

**Theorem 5.4** (Seasonal Attention Pattern from Periodic Keys and Dominant RoPE Channel). *Suppose the query and key vectors are approximately periodic with interval L, in the sense that*

$$\|q_{t+L} - q_t\| \le \varepsilon_q, \qquad \|k_{i+L} - k_i\| \le \varepsilon_k$$

*for sufficiently small $\varepsilon_q, \varepsilon_k > 0$, and that this interval is in near resonance with the dominant RoPE frequency, i.e., $\left|L\,\theta_{m^\star} - 2k\pi\right| \le \delta$ for some positive integer k and sufficiently small $\delta > 0$. Then the attention logits satisfy*

$$|a_{t+L,i} - a_{t,i}| \le C_1(\varepsilon_q + \varepsilon_k) + C_2\delta, \quad |a_{t,i+L} - a_{t,i}| \le C_3(\varepsilon_q + \varepsilon_k) + C_4\delta$$

*for some constants $C_1, C_2, C_3, C_4 > 0$, and therefore exhibit a* seasonal pattern *with period L along both query and key dimensions.*

The seasonal pattern arises from two combined effects. First, the approximate periodicity of the input queries and keys induces a corresponding periodicity in the attention map. This type of periodicity is common in structured data, such as looking at corresponding elements in consecutive lines of code or data records. Second, when the interval $L$ is in resonance with the dominant RoPE frequency, the relative-position rotations align with the input periodicity, reinforcing the repetition and producing a stronger, more regular pattern. This dual condition—periodic keys amplified by RoPE resonance—explains the emergence of clean, regularly spaced attention pattern. The observed interval $L$ is therefore determined primarily by the period of the input data itself.

# 6 DOWNSTREAM TASKS

## 6.1 KV CACHE COMPRESSION

To demonstrate the practical value of TAPPA, we apply *q-similarity*, a metric derived from TAPPA, to the **KV cache compression** task, which aims to reduce the memory footprint of key-value caches during large language model inference while maintaining model accuracy. Based on TAPPA, lower

Table 2: Comparison of TAPPA with ShortGPT under the same pruning ratios.

| Model | Method | Pruned | Piqa | Hellaswag | Winogrande | Arc Easy | Average (%)↑ |
|-------|--------|--------|------|-----------|-----------|----------|--------------|
| **Llama-2-7B** | ShortGPT | 31% | 63.33 | 45.94 | 61.40 | **47.26** | 54.48 |
| | ∼ with TAPPA | 31% | **63.87** | **50.83** | **63.54** | 45.03 | **55.82** |
| | ShortGPT | 34% | 60.83 | 42.11 | 60.38 | **44.15** | 51.87 |
| | ∼ with TAPPA | 34% | 60.45 | **48.53** | **62.43** | 42.55 | **53.49** |
| **Llama-3.1-8B** | ShortGPT | 28% | **66.65** | 42.41 | 58.72 | 46.25 | 53.51 |
| | ∼ with TAPPA | 28% | 64.69 | **55.09** | **63.77** | **52.90** | **59.11** |
| | ShortGPT | 31% | 64.96 | 37.69 | 58.41 | 42.76 | 50.96 |
| | ∼ with TAPPA | 31% | **65.51** | **42.22** | **62.51** | **46.59** | **54.21** |
| **Qwen-2.5-7B** | ShortGPT | 39% | **63.17** | **41.83** | 50.59 | 44.32 | 49.98 |
| | ∼ with TAPPA | 39% | 62.89 | 41.80 | **51.93** | **45.03** | **50.42** |
| | ShortGPT | 43% | 60.83 | 36.13 | 47.43 | 39.77 | 46.04 |
| | ∼ with TAPPA | 43% | **60.88** | **39.87** | **49.72** | **43.94** | **48.60** |

query similarity indicates a higher likelihood of retrieval patterns. Since retrieval patterns attend to scattered and unpredictable key positions, they generally require a larger cache budget to preserve critical information (Xiao et al., 2024; Li et al., 2025). Therefore, we leverage q-similarity as a proxy signal to dynamically guide the per-layer cache budget allocation under limited memory resources, improving inference efficiency while maintaining model accuracy. We provide the experiment details in Appendix G.1, and additional studies on the sensitivity to $\alpha$ and alternative similarity formulations for q-similarity are reported in Appendix H.

**Results.** As shown in Table 1, our method consistently outperforms CAKE and the other four baselines across three different budget settings. These results confirm that *q-similarity* derived from TAPPA effectively reflects the likelihood of retrieval patterns, and by allocating more cache budget to layers exhibiting lower query similarity, we are able to preserve critical information more effectively, enabling efficient KV cache compression. More results comparing with DuoAttention (Xiao et al., 2024) and Expected Attention (Devoto et al., 2025) are in Appendix I and J.

## 6.2 LLM PRUNING

To reduce the parameter size of LLMs and accelerate inference, structured pruning, which removes entire components such as layers, has emerged as a promising approach. Our specific goal is to design more effective proxy metrics to guide whole-layer pruning, so as to achieve higher accuracy under the same compression ratio. Based on TAPPA, higher q-similarity indicates more stable and predictable patterns. Such stability suggests that the layer extracts less novel information, making it more dispensable. Consequently, layers with higher query similarity can be pruned with less impact on model performance, while low similarity layers, which are more likely to host retrieval-oriented and task-critical behaviors, are preserved. We provide the experiment details in Appendix G.2, and the sensitivity study of $\beta$ is reported in Appendix H.

**Results.** As shown in Table 2, our method consistently outperforms ShortGPT across different pruning ratios and models, validating the effectiveness of combining Block Influence with q-similarity as a proxy signal for structured layer pruning. These results on LLM pruning validate our hypothesis regarding the connection between q-similarity and stable, predictable patterns. Layers with higher q-similarity exhibit greater redundancy due to their stability, and can therefore be pruned with minimal impact on overall model performance. Results about more baselines are in Appendix G.2.1.

## 7 CONCLUSION

In this work, we introduced TAPPA, a unifying framework to systematically analyze the diverse attention patterns within large language models. We demonstrated that the distinction between predictable and unpredictable patterns can be explained by the temporal self-similarity of queries. Our theoretical analysis further elucidated that stable, predictable patterns arise from the combined effects of query-key continuity and Rotary Positional Embeddings (RoPE), providing a clear explanation for phenomena like periodic sequential diagonals. The practical value of TAPPA is confirmed by applying its insights to downstream tasks. A simple metric inspired by TAPPA successfully improved performance in both KV cache compression and LLM pruning, validating our framework.

## ACKNOWLEDGMENTS

This work was supported in part by National Key R&D Program of China under contract 2022ZD0119801, National Nature Science Foundations of China grants U23A20388, 62021001, U19B2026, and U19B2044. This work was supported in part by Huawei as well. We would like to thank all the anonymous reviewers for their insightful comments. This research was also supported by the advanced computing resources provided by the Supercomputing Center of the USTC.

## ETHICS STATEMENT

This research does not involve any personally identifiable information. All datasets used are publicly available and widely adopted in the community, and we have verified that their licenses permit research use. In accordance with the ICLR Code of Ethics, we ensure that our work adheres to principles of fairness, transparency, and responsible AI research. We also disclose that LLMs were used for text polishing, while all conceptual contributions and validation remain the responsibility of the authors in Appendix L.

## REPRODUCIBILITY STATEMENT

We will provide open access to all source code, configuration files, and preprocessing scripts, together with detailed instructions to reproduce the main experimental results. All datasets employed are publicly available, and we specify the exact versions and preprocessing steps. Collectively, these resources and specifications enable reliable and faithful reproduction of our results.

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

## A  PROOF OF UNPREDICTABLE PATTERN

**Proposition 4.1.** *Let $q_t, q_{t+1} \in \mathbb{R}^d$ be consecutive queries, $K = [k_1, \ldots, k_T]^\top$ the key matrix, and define the logits*

$$a_{t,j} = q_t^\top R_{t-j} k_j, \qquad a_{t+1,j} = q_{t+1}^\top R_{t+1-j} k_j.$$

*If $q_{t+1} - q_t$ has a large norm and is not orthogonal to all rotated keys $\{R_{t+1-j}k_j\}$, then the difference between the logit vectors $a_t$ and $a_{t+1}$ is necessarily large. In particular, there exist constants $c_1, c_2 > 0$ such that*

$$\|a_{t+1} - a_t\|_\infty \geq c_1 \|q_{t+1} - q_t\| - c_2.$$

*Proof.* Let $\Delta q := q_{t+1} - q_t$. For each position $j$, the change in the logit is

$$\Delta a_j = a_{t+1,j} - a_{t,j} = (\Delta q)^\top R_{t+1-j} k_j + q_t^\top (R_{t+1-j} - R_{t-j}) k_j.$$

Denote the first term by $T_{1,j}$ and the second by $T_{2,j}$.

**Step 1: Bounding the RoPE difference term.** Since $R_m$ is an orthogonal rotation, its operator norm is 1, so by the triangle inequality, $\|R_{t+1-j} - R_{t-j}\|_{\mathrm{op}} \leq \|R_{t+1-j}\|_{\mathrm{op}} + \| - R_{t-j}\|_{\mathrm{op}} \leq 2$. If we assume the keys are bounded such that $\|k_j\| \leq B_K$ for all $j$, then

$$|T_{2,j}| \leq \|q_t\| \, \|R_{t+1-j} - R_{t-j}\|_{\mathrm{op}} \, \|k_j\| \leq 2 \|q_t\| B_K.$$

**Step 2: Lower-bounding the query difference term.** The first term can be written as

$$|T_{1,j}| = \|\Delta q\| \cdot \left| \langle \tfrac{\Delta q}{\|\Delta q\|}, R_{t+1-j} k_j \rangle \right|.$$

The condition that $\Delta q$ is not orthogonal to all rotated keys implies that the inner product is not always zero. We formalize this by assuming there exists an index $j^*$ and a constant $\alpha > 0$ such that the normalized vectors have a significant projection:

$$\left| \langle \tfrac{\Delta q}{\|\Delta q\|}, R_{t+1-j^*} k_{j^*} / \|k_{j^*}\| \rangle \right| \geq \alpha.$$

This condition essentially states that the direction of the query change aligns with at least one rotated key. Under this condition, and assuming a minimum key norm $\|k_{j^*}\| \geq B_{k,\min}$, we get

$$|T_{1,j^*}| \geq \alpha B_{k,\min} \|\Delta q\|.$$

**Step 3: Combining both terms.** Using the bounds for the two terms at index $j^*$, the reverse triangle inequality gives

$$|\Delta a_{j^*}| \geq |T_{1,j^*}| - |T_{2,j^*}| \geq \alpha B_{k,\min} \|\Delta q\| - 2 \|q_t\| B_K.$$

Since the infinity norm of a vector is the maximum of the absolute values of its components, we have

$$\|a_{t+1} - a_t\|_\infty = \max_j |\Delta a_j| \geq |\Delta a_{j^*}| \geq \alpha B_{k,\min} \|\Delta q\| - 2 \|q_t\| B_K.$$

This establishes the proposition with constants $c_1 = \alpha B_{k,\min}$ and $c_2 = 2 \|q_t\| B_K$. This completes the proof. $\square$

## B  PROOF OF RE-ACCESS PATTERN

**Theorem 5.1**(Vertical Stability of Attention): *Suppose the channel-wise decomposition (Eq. 5) holds for the attention logits $a_{t,i}$. Assume that the queries evolve continuously in the sense that $\|q_{t+1} - q_t\| \leq \varepsilon$, while all keys $k_i$ remain fixed between steps $t$ and $t + 1$. Further assume the existence of a dominant low-frequency channel $m^\star$ whose weight $w_{m^\star}$ dominates the other channels, and whose RoPE frequency $\theta_{m^\star}$ is small. Then the per-key differences $a_{t+1,i} - a_{t,i}$ are uniformly small, and the attention logits are vertically stable.*

*Proof.* We derive an explicit uniform bound for the per-key logit difference and show how it depends on the query increment and channel parameters.

Using the channel decomposition from Eq. 5, write for each channel $m$

$$w_m := \|q_t^{(m)}\| \, \|k_i^{(m)}\|, \qquad w_m' := \|q_{t+1}^{(m)}\| \, \|k_i^{(m)}\|,$$

and

$$\psi_m := \phi_{t,i}^{(m)} + (i-t)\theta_m, \qquad \psi_m' := \phi_{t+1,i}^{(m)} + (i-(t+1))\theta_m.$$

Define the logit difference

$$\Delta_{t,i} := a_{t+1,i} - a_{t,i}.$$

Direct subtraction yields the exact identity

$$\Delta_{t,i} = \sum_{m=1}^{M} (w_m' - w_m) \cos \psi_m' \; + \; \sum_{m=1}^{M} w_m \big( \cos \psi_m' - \cos \psi_m \big). \tag{8}$$

We bound the two sums on the right-hand side separately. Let

$$\varepsilon := \|q_{t+1} - q_t\|.$$

**First sum.** By the triangle inequality and the definition of $w_m$,

$$\Big| \sum_{m=1}^{M} (w_m' - w_m) \cos \psi_m' \Big| \le \sum_{m=1}^{M} |w_m' - w_m| = \sum_{m=1}^{M} \|k_i^{(m)}\| \, \big| \|q_{t+1}^{(m)}\| - \|q_t^{(m)}\| \big|.$$

Since the Euclidean norm is 1-Lipchitz,

$$\big| \|q_{t+1}^{(m)}\| - \|q_t^{(m)}\| \big| \le \|q_{t+1}^{(m)} - q_t^{(m)}\| \le \|q_{t+1} - q_t\| = \varepsilon.$$

Hence

$$\Big| \sum_{m=1}^{M} (w_m' - w_m) \cos \psi_m' \Big| \le \varepsilon \sum_{m=1}^{M} \|k_i^{(m)}\|. \tag{9}$$

**Second sum.** Use the inequality $|\cos u - \cos v| \le |u - v|$, so

$$\big| \cos \psi_m' - \cos \psi_m \big| \le |\psi_m' - \psi_m| = \big| \phi_{t+1,i}^{(m)} - \phi_{t,i}^{(m)} - \theta_m \big| \le |\phi_{t+1,i}^{(m)} - \phi_{t,i}^{(m)}| + |\theta_m|.$$

To control the angular difference, let $r_m := \min\{\|q_t^{(m)}\|, \|q_{t+1}^{(m)}\|\}$ and assume $r_m > 0$, and denote $\varepsilon^{(m)} := \|q_{t+1}^{(m)} - q_t^{(m)}\|$. In the 2D RoPE subspace, write $q_t^{(m)} = r_t u_t$ and $q_{t+1}^{(m)} = r_{t+1} u_{t+1}$ with $\|u_t\| = \|u_{t+1}\| = 1$ and let $\Delta\phi^{(m)} := \phi_{t+1,i}^{(m)} - \phi_{t,i}^{(m)}$ be the angle between $q_t^{(m)}$ and $q_{t+1}^{(m)}$. Projecting both vectors onto the circle of radius $r_m$ can only decrease their Euclidean distance while preserving the angle, so by elementary planar geometry, we have

$$2 r_m \sin\left( \frac{|\Delta\phi^{(m)}|}{2} \right) \; \le \; \varepsilon^{(m)}.$$

Therefore

$$|\phi_{t+1,i}^{(m)} - \phi_{t,i}^{(m)}| = |\Delta\phi^{(m)}| \le 2 \arcsin\left( \frac{\varepsilon^{(m)}}{2 r_m} \right) \le \frac{\pi}{2} \frac{\varepsilon^{(m)}}{r_m} \le \frac{\pi}{2} \frac{\varepsilon}{r_m},$$

where we used $\varepsilon^{(m)} \le \|q_{t+1} - q_t\| = \varepsilon$ in the last inequality.

Therefore

$$\big| w_m \big( \cos \psi_m' - \cos \psi_m \big) \big| \le w_m \Big( \frac{\pi}{2} \frac{\varepsilon}{r_m} + |\theta_m| \Big).$$

Summing over $m$ yields

$$\Big| \sum_{m=1}^{M} w_m \big( \cos \psi_m' - \cos \psi_m \big) \Big| \le \frac{\pi}{2} \varepsilon \sum_{m=1}^{M} \frac{w_m}{r_m} \; + \; \sum_{m=1}^{M} w_m |\theta_m|. \tag{10}$$

**Combine bounds.** Inserting equation 9 and equation 10 into equation 8 gives the explicit uniform bound

$$|\Delta_{t,i}| \le \varepsilon \sum_{m=1}^{M} \|k_i^{(m)}\| \;+\; \frac{\pi}{2}\,\varepsilon \sum_{m=1}^{M} \frac{w_m}{r_m} \;+\; \sum_{m=1}^{M} w_m |\theta_m|. \tag{11}$$

Define

$$\delta := \varepsilon \sum_{m=1}^{M} \|k_i^{(m)}\| \;+\; \frac{\pi}{2}\,\varepsilon \sum_{m=1}^{M} \frac{w_m}{r_m} \;+\; \sum_{m=1}^{M} w_m |\theta_m|.$$

Thus $|\Delta_{t,i}| \le \delta$ for every token index $i$.

**Conclusion and asymptotics.** Under the theorem hypotheses the keys are bounded and there exists a dominant channel $m^\star$ with $w_{m^\star}$ much larger than the remaining $\{w_m\}_{m \ne m^\star}$, while $r_{m^\star}$ is bounded away from zero and $|\theta_{m^\star}|$ is small. In that regime, the two terms proportional to $\varepsilon$ in $\delta$ vanish as $\varepsilon \to 0$, and the last term is small because the dominant channel's frequency $|\theta_{m^\star}|$ is small and the remaining channels carry only a small total weight. Consequently $\delta$ can be made arbitrarily small by taking $\varepsilon \to 0$, $|\theta_{m^\star}| \to 0$, and by increasing the dominance of $w_{m^\star}$ over other channel weights. Therefore the per-key differences $\Delta_{t,i} = a_{t+1,i} - a_{t,i}$ are uniformly small, which proves vertical stability. $\qquad\square$

## C    PROOF OF SEQUENTIAL PATTERN

**Theorem 5.2**(Sequential Patterns under High Self-similarity): *Under the RoPE relative-position encoding, suppose queries and keys both exhibit high self-similarity, in the sense that*

$$\|q_{t+1} - q_t\| \le \varepsilon, \qquad \|k_{i+1} - k_i\| \le \varepsilon$$

*for sufficiently small $\varepsilon > 0$. Then the attention logits satisfy*

$$|a_{t+1,i+1} - a_{t,i}| \le C\varepsilon,$$

*for some constant $C > 0$. Consequently, the attention logits exhibit approximate shift-invariance along the $(+1,+1)$ diagonal, giving rise to sequential patterns in the attention map.*

*Proof.* Recall the attention logit

$$a_{t,i} := q_t^\top R_{t-i} k_i,$$

where $R_\Delta$ is the RoPE rotation for relative offset $\Delta$. By the RoPE identity we have $R_{(t+1)-(i+1)} = R_{t-i}$, hence

$$a_{t+1,i+1} = q_{t+1}^\top R_{t-i} k_{i+1}.$$

Therefore, the difference can be written as

$$a_{t+1,i+1} - a_{t,i} = (q_{t+1} - q_t)^\top R_{t-i} k_{i+1} + q_t^\top R_{t-i}(k_{i+1} - k_i).$$

Taking absolute values and applying the Cauchy–Schwarz inequality gives

$$\begin{aligned} \left| a_{t+1,i+1} - a_{t,i} \right| &\le \|q_{t+1} - q_t\| \, \|R_{t-i} k_{i+1}\| \;+\; \|q_t\| \, \|R_{t-i}(k_{i+1} - k_i)\| \\ &= \|q_{t+1} - q_t\| \, \|k_{i+1}\| \;+\; \|q_t\| \, \|k_{i+1} - k_i\|, \end{aligned}$$

where the last equality uses that each $R_\Delta$ is orthogonal (rotation), hence $\|R_\Delta v\| = \|v\|$.

Now impose the high self-similarity hypothesis in the rigorous form

$$\|q_{t+1} - q_t\| \le \varepsilon, \qquad \|k_{i+1} - k_i\| \le \varepsilon$$

for some $\varepsilon > 0$. Further assume the query/key vectors are uniformly norm-bounded, i.e. there exist constants $Q, K > 0$ with $\|q_t\| \le Q$ and $\|k_i\| \le K$ for all relevant $t, i$. Then

$$\left| a_{t+1,i+1} - a_{t,i} \right| \le \varepsilon \, \|k_{i+1}\| + \|q_t\|\,\varepsilon \le \varepsilon(K + Q).$$

Setting $C := K + Q$ yields the claimed bound

$$\left| a_{t+1,i+1} - a_{t,i} \right| \le C\,\varepsilon.$$

Thus, under the stated assumptions, the attention logits are approximately shift-invariant along the $(+1,+1)$ diagonal (with error at most $C\varepsilon$), which produces the sequential diagonal structure in the logit map. $\qquad\square$

## D  PROOF OF PERIODIC SEQUENTIAL PATTERN

**Theorem 5.3** (Periodic Sequential Pattern from a Dominant RoPE Channel): *If a sequential pattern arises and the corresponding key exhibits a massive channel at index $m^\star$, then the spacing between adjacent diagonals is determined by the rotation frequency of that channel:*

$$T \;=\; \frac{2\pi}{\theta_{m^\star}} \;=\; 2\pi\, c^{2m^\star/d}. \tag{12}$$

*Proof.* From the decomposition view of attention, the attention logits can be written as a sum over channels:

$$a_{t,i} = \sum_{m=1}^{M} \|q_t^{(m)}\| \, \|k_i^{(m)}\| \cos\big(\phi_{t,i}^{(m)} + (i-t)\theta_m\big).$$

By assumption, channel $m^\star$ is massive, meaning its contribution to $a_{t,i}$ dominates all other channels:

$$\|q_t^{(m^\star)}\| \, \|k_i^{(m^\star)}\| \gg \|q_t^{(m)}\| \, \|k_i^{(m)}\| \quad \text{for all } m \neq m^\star.$$

Hence, the logits are approximately

$$a_{t,i} \approx \|q_t^{(m^\star)}\| \, \|k_i^{(m^\star)}\| \cos\big(\phi_{t,i}^{(m^\star)} + (i-t)\theta_{m^\star}\big).$$

Consider positions $i$ and $i+T$. Assuming that the magnitudes $\|k_i^{(m^\star)}\|$ and angles $\phi_{t,i}^{(m^\star)}$ vary slowly across consecutive tokens forming the sequential pattern, the attention pattern repeats whenever

$$(i-t)\theta_{m^\star} \;\equiv\; (i+T-t)\theta_{m^\star} \pmod{2\pi},$$

which yields

$$T = \frac{2\pi}{\theta_{m^\star}}.$$

By the definition of RoPE, $\theta_m = c^{-2m/d}$, and substituting $m = m^\star$ gives

$$T = \frac{2\pi}{\theta_{m^\star}} = 2\pi\, c^{2m^\star/d}.$$

Therefore, the interval between adjacent diagonals in the attention map is exactly determined by the rotation frequency of the dominant channel, as claimed. $\square$

## E  PROOF OF SEASONAL PATTERN

**Theorem 5.4** (Seasonal Attention Pattern from Periodic Keys and Dominant RoPE Channel): *Suppose the query and key vectors are approximately periodic with interval $L$, in the sense that*

$$\|q_{t+L} - q_t\| \leq \varepsilon_q, \qquad \|k_{i+L} - k_i\| \leq \varepsilon_k$$

*for sufficiently small $\varepsilon_q, \varepsilon_k > 0$, and that this interval is in near resonance with the dominant RoPE frequency, i.e.,*

$$\big| L\,\theta_{m^\star} - 2k\pi \big| \leq \delta$$

*for some positive integer $k$ and sufficiently small $\delta > 0$. Then the attention logits satisfy*

$$|a_{t+L,i} - a_{t,i}| \leq C_1(\varepsilon_q + \varepsilon_k) + C_2\delta, \quad |a_{t,i+L} - a_{t,i}| \leq C_3(\varepsilon_q + \varepsilon_k) + C_4\delta$$

*for some constants $C_1, C_2, C_3, C_4 > 0$, and therefore exhibit a* seasonal pattern *with period $L$ along both query and key dimensions.*

**Proof.** We again use the channel-wise RoPE decomposition. For each channel $m$, let $R_t^{(m)}$ and $R_i^{(m)}$ denote the $2 \times 2$ rotation matrices induced by RoPE at positions $t$ and $i$ with angular frequency $\theta_m$. We define the post-RoPE query and key components as

$$\tilde{q}_t^{(m)} := R_t^{(m)} q_t^{(m)}, \qquad \tilde{k}_i^{(m)} := R_i^{(m)} k_i^{(m)}.$$

By construction, RoPE is an orthogonal transformation, so $\|\tilde{q}_t^{(m)}\| = \|q_t^{(m)}\|$ and $\|\tilde{k}_i^{(m)}\| = \|k_i^{(m)}\|$. The logit contributed by channel $m$ can be written as a dot product

$$a_{t,i}^{(m)} = \langle \tilde{q}_t^{(m)}, \tilde{k}_i^{(m)} \rangle, \quad a_{t,i} = \sum_{m=1}^{M} a_{t,i}^{(m)}.$$

We first bound the variation of the *dominant* channel $m^\star$ along the query dimension. For arbitrary vectors $u, u', v, v'$, we use the standard dot-product inequality

$$|u^\top v - u'^\top v'| \le \|v\| \, \|u - u'\| + \|u'\| \, \|v - v'\|. \tag{$\star$}$$

Applying $(\star)$ with $u = \tilde{q}_{t+L}^{(m^\star)}$, $u' = \tilde{q}_t^{(m^\star)}$ and $v = v' = \tilde{k}_i^{(m^\star)}$ gives

$$|a_{t+L,i}^{(m^\star)} - a_{t,i}^{(m^\star)}| = |\langle \tilde{q}_{t+L}^{(m^\star)}, \tilde{k}_i^{(m^\star)} \rangle - \langle \tilde{q}_t^{(m^\star)}, \tilde{k}_i^{(m^\star)} \rangle|$$
$$\le \|\tilde{k}_i^{(m^\star)}\| \, \|\tilde{q}_{t+L}^{(m^\star)} - \tilde{q}_t^{(m^\star)}\|. \tag{13}$$

It remains to control $\|\tilde{q}_{t+L}^{(m^\star)} - \tilde{q}_t^{(m^\star)}\|$. Using the definition of $\tilde{q}_t^{(m^\star)}$ we have

$$\tilde{q}_{t+L}^{(m^\star)} - \tilde{q}_t^{(m^\star)} = R_{t+L}^{(m^\star)} q_{t+L}^{(m^\star)} - R_t^{(m^\star)} q_t^{(m^\star)}$$
$$= R_{t+L}^{(m^\star)} \big( q_{t+L}^{(m^\star)} - q_t^{(m^\star)} \big) + \big( R_{t+L}^{(m^\star)} - R_t^{(m^\star)} \big) q_t^{(m^\star)}. \tag{14}$$

Taking norms and using orthogonality of $R_{t+L}^{(m^\star)}$ yields

$$\|\tilde{q}_{t+L}^{(m^\star)} - \tilde{q}_t^{(m^\star)}\| \le \|q_{t+L}^{(m^\star)} - q_t^{(m^\star)}\| + \big\| \big( R_{t+L}^{(m^\star)} - R_t^{(m^\star)} \big) q_t^{(m^\star)} \big\|. \tag{15}$$

The first term is controlled by the assumed $L$-periodicity of the queries:

$$\|q_{t+L}^{(m^\star)} - q_t^{(m^\star)}\| \le \varepsilon_q.$$

For the second term, we use the near-resonance condition. By the definition of RoPE, $R_{t+L}^{(m^\star)} = R_t^{(m^\star)} R_L^{(m^\star)}$, where $R_L^{(m^\star)}$ is a rotation by angle $L\theta_{m^\star}$ in the channel-$m^\star$ plane. The hypothesis $|L\theta_{m^\star} - 2k\pi| \le \delta$ means that $R_L^{(m^\star)}$ is in fact a rotation by an angle of magnitude at most $\delta$ around the identity. For a planar rotation by angle $\gamma$, we have $\|R(\gamma) - I\| = 2|\sin(\gamma/2)| \le |\gamma|$, so

$$\big\| \big( R_{t+L}^{(m^\star)} - R_t^{(m^\star)} \big) q_t^{(m^\star)} \big\| = \big\| R_t^{(m^\star)} \big( R_L^{(m^\star)} - I \big) q_t^{(m^\star)} \big\|$$
$$\le \|R_L^{(m^\star)} - I\| \, \|q_t^{(m^\star)}\| \le \delta \, \|q_t^{(m^\star)}\|. \tag{16}$$

Combining equation 15 and equation 16 gives

$$\|\tilde{q}_{t+L}^{(m^\star)} - \tilde{q}_t^{(m^\star)}\| \le \varepsilon_q + \delta \, \|q_t^{(m^\star)}\|.$$

Substituting this into equation 13 and recalling $\|\tilde{k}_i^{(m^\star)}\| = \|k_i^{(m^\star)}\|$ yields

$$|a_{t+L,i}^{(m^\star)} - a_{t,i}^{(m^\star)}| \le \|k_i^{(m^\star)}\| \varepsilon_q + \|k_i^{(m^\star)}\| \|q_t^{(m^\star)}\| \delta =: C_1^{(\star)} \varepsilon_q + C_2^{(\star)} \delta.$$

An entirely symmetric argument, exchanging the roles of $t$ and $i$ and using the $L$-periodicity of the keys $\|k_{i+L}^{(m^\star)} - k_i^{(m^\star)}\| \le \varepsilon_k$, shows that

$$|a_{t,i+L}^{(m^\star)} - a_{t,i}^{(m^\star)}| \le C_3^{(\star)} \varepsilon_k + C_4^{(\star)} \delta$$

for some constants $C_3^{(\star)}, C_4^{(\star)} > 0$ depending only on the norms of $q_t^{(m^\star)}$ and $k_i^{(m^\star)}$.

Finally, recall that channel $m^\star$ is assumed to be *massive*: its contribution $\|q_t^{(m^\star)}\|\|k_i^{(m^\star)}\|$ dominates the contributions of all other channels. The residual variation coming from non-dominant channels $\{m \ne m^\star\}$ is therefore uniformly bounded and can be absorbed into the constants $C_1, \dots, C_4$. Renaming the constants and noting that $\varepsilon_q + \varepsilon_k \ge \varepsilon_q$ and $\varepsilon_q + \varepsilon_k \ge \varepsilon_k$, we obtain the bounds stated in Theorem 5.4:

$$|a_{t+L,i} - a_{t,i}| \le C_1(\varepsilon_q + \varepsilon_k) + C_2\delta, \quad |a_{t,i+L} - a_{t,i}| \le C_3(\varepsilon_q + \varepsilon_k) + C_4\delta.$$

This shows that the dominant component of the attention logits approximately repeats every $L$ steps along both query and key dimensions, giving rise to a seasonal pattern with period $L$. $\qquad \square$

# F EMPIRICAL SUPPORT

## F.1 EMPIRICAL VALIDATION OF THE DOMINANT-CHANNEL ASSUMPTION OF RE-ACCESS PATTERN

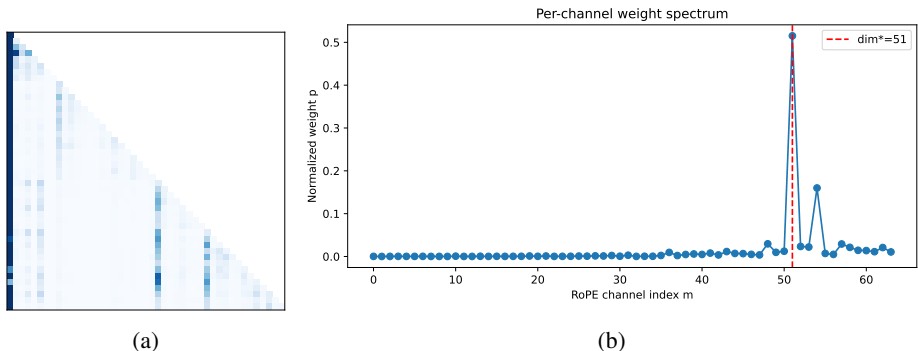

Figure 6: Empirical validation of the dominant-channel assumption for a re-access head. (a) is an attention heatmap of the re-access pattern. (b) plots the RoPE-channel weights of attention at the sink position (dark vertical stripe), showing that a single low-frequency channel $m^*$ accounts for most of the total weight.

Theorem 5.1 assumes that the attention logits of re-access heads are dominated by a single low-frequency channel. To directly examine this assumption, we perform a simple spectrum analysis on a head whose attention map exhibits a clear re-access pattern in Figure 6(a).

For this head, we focus on the key position corresponding to the re-access stripe, namely the attention sink. We decompose the query and key vectors into $M = D/2$ RoPE channels, where each channel $m$ groups the two feature dimensions that share the same RoPE frequency. For every channel $m$, we aggregate its contribution over the decoding steps and then normalize the resulting values so that they sum to 1. This gives a one-dimensional spectrum $\{p_m\}_{m=0}^{M-1}$.

Figure 6(b) plots the weight of each attention channel. The horizontal axis is the RoPE channel index $m$ ($0 \leq m < M$), and the vertical axis is the *normalized channel weight* $p_m$, i.e., the relative contribution of each channel to the attention logits at the sink position. We observe a highly concentrated pattern: a single channel $m^*$ carries about $p_{m^*} \approx 51\%$ of the total mass, while the remaining channels form a long tail with much smaller weights. The dominant channel $m^*$ lies in the low-frequency half of the RoPE spectrum, consistent with the dominant low-frequency channel assumption used in Theorem 5.1.

Together, these observations provide direct empirical evidence that, for the re-access heads we analyze, the attention logits are indeed governed by a single low-frequency channel.

## F.2 DISENTANGLING QUERY DYNAMICS AND RoPE IN SEQUENTIAL PATTERN.

To empirically separate the roles of input dynamics and RoPE, we conduct a controlled ablation on a single attention head that exhibits a clear sequential pattern. For this head, the average cosine similarity between consecutive queries is approximately 0.99, and the full model (with RoPE enabled) produces an almost perfectly smooth diagonal attention pattern.

We construct three variants using the same head and the same input sequence (Figure 7):

1. **High q-similarity with RoPE (full model).** In the original model, both the queries and keys have high temporal self-similarity, and RoPE is applied as usual. The resulting attention map shows a clean, nearly translation-invariant diagonal stripe: as $t$ increases, the high-attention region shifts along the $(+1, +1)$ direction with very little distortion. This behavior is consistent with the analysis of TAPPA, which predicts that when both $q_t$ and $k_i$ vary smoothly in time, RoPE induces approximate shift-invariance along the main diagonal.

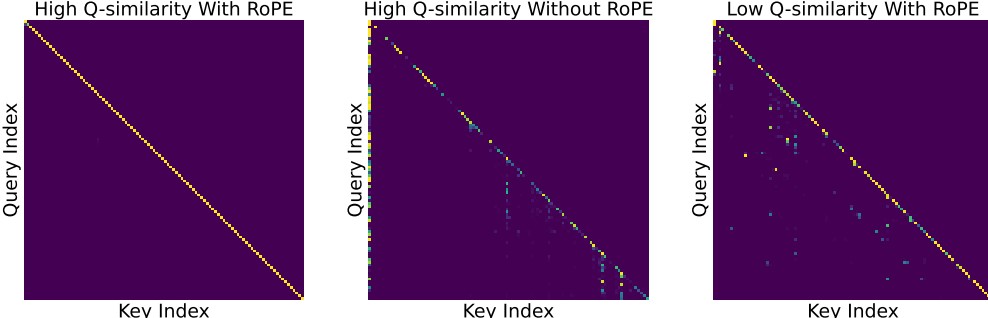

Figure 7: Ablation of query dynamics and RoPE on a head with a strong sequential pattern. **Left**: original head with high q-similarity and RoPE enabled. **Middle**: high q-similarity without RoPE, which retains a rough, broken diagonal with additional vertical streaks. **Right**: RoPE with perturbed q, where the diagonal tendency is overlaid with scattered, unpredictable activation spikes.

2. **High q-similarity without RoPE.** In the second variant, we disable RoPE for this head by replacing the rotation matrices with identity, while keeping the original queries and keys unchanged. The attention map still exhibits a diagonal bias, reflecting the strong local similarity in the queries and keys. However, the diagonal becomes noticeably rough: it is broken into segments and is superposed with vertical streaks. This indicates that high q-similarity alone is sufficient to encourage local, near-diagonal attention, but it does not guarantee the smooth, globally shift-invariant diagonal pattern observed in the full model.

3. **Perturbed q-dynamics with RoPE.** In the third variant, we keep RoPE enabled but mildly perturb the temporal dynamics of the queries by randomly resampling their time indices within the same sequence. This reduces the average cosine similarity between consecutive queries from 0.99 to 0.97, while leaving the keys and RoPE parameters unchanged. The resulting attention map still contains a visible diagonal tendency, but it is now overlaid with many scattered, seemingly random activation spots. In other words, the attention pattern becomes a mixture of a predictable diagonal component and unpredictable spikes.

Across these three conditions, we observe that: (i) high q-similarity without RoPE yields a coarse, locally diagonal pattern, (ii) RoPE with perturbed q-dynamics produces partially diagonal but noticeably more unpredictable attention, and (iii) only when smooth q-dynamics and RoPE are both present do we obtain the clean, stable sequential pattern seen in the full model. This ablation supports the mechanism identified by TAPPA that sequential attention patterns arise from the *joint* effect of smooth input dynamics and RoPE, and that these two factors play complementary roles: input dynamics control whether the pattern is predictable or unpredictable, while RoPE shapes the predictable component into a regular, shift-invariant diagonal structure.

### F.3 Q-SIMILARITY DISTRIBUTION

To better understand the behavior of q-similarity, we compute per-head q-similarity scores across all layers for two models (Llama-3.1 and Qwen2.5-7B) on two representative datasets (GSM8K and AIGC). As shown in Figure 8, we have following observations:

**Overall high q-similarity supporting temporal continuity.** Across all heads and layers, the average q-similarity is high for both models (around 0.80 for Llama-3.1 and 0.86 for Qwen2.5-7B). This empirically supports the assumption of TAPPA that queries tend to evolve in a temporally continuous manner in a large portion of the network.

**Model-specific but layer-structured distributions.** Each model exhibits its own characteristic distribution of q-similarity values, indicating that the q-similarity distribution reflects model-specific properties and thus naturally calls for per-model calibration. At the same time, within a given model, we observe a clear and consistent structure: heads in the *same* layer have very similar q-similarity scores (forming tight clusters), whereas the average q-similarity differs significantly *across* layers.

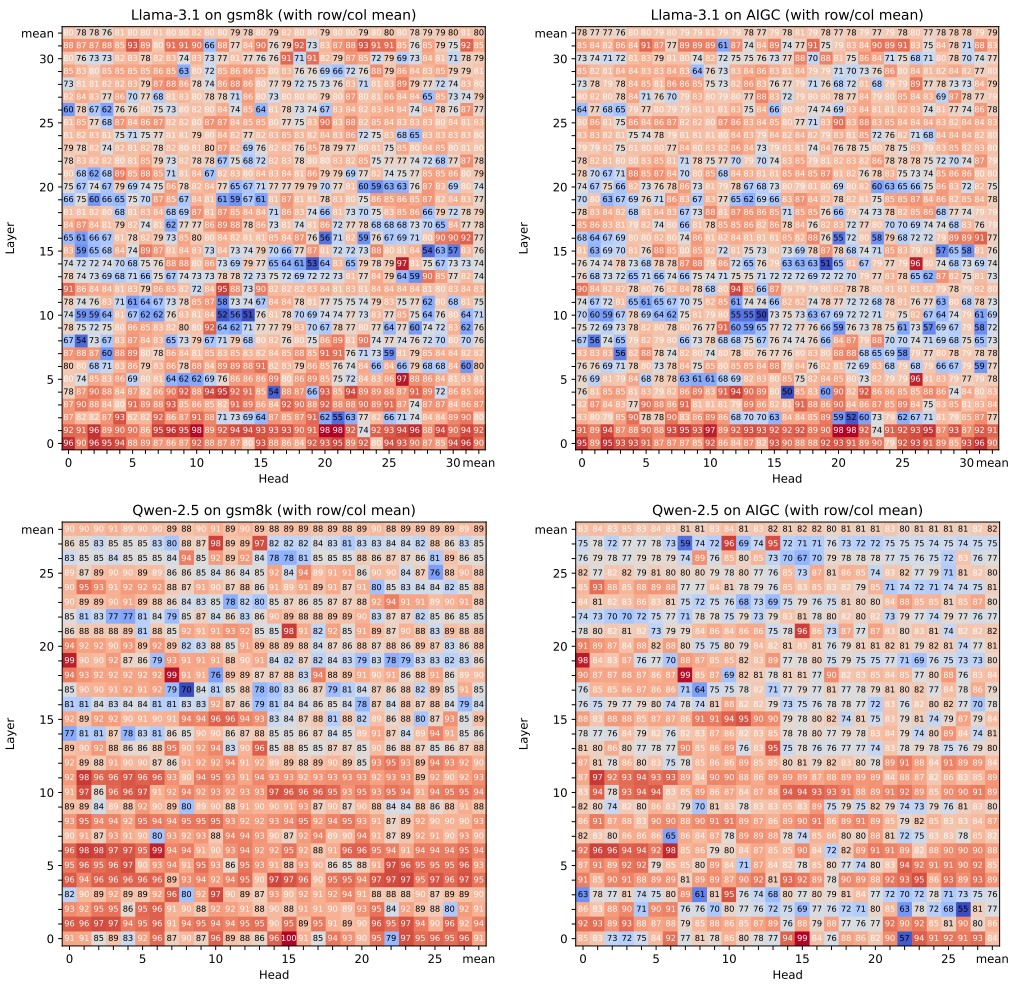

Figure 8: Head-wise q-similarity heatmaps for Llama-3.1 and Qwen2.5 on GSM8K and AIGC. For readability, we show only the two decimal digits of each q-similarity value (e.g., "83" denotes a q-similarity of 0.83).

This justifies the design choice in TAPPA of operating at the layer level by using layer-wise averages when building downstream metrics and policies.

**Stable across datasets for the same model, enabling lightweight calibration.** For a fixed model, the q-similarity distribution is highly consistent across datasets. For Llama-3.1, the average q-similarity on GSM8K and AIGC differs by only about 0.01. For Qwen2.5-7B, the absolute mean difference between the two datasets is about 0.07, but the overall shape and ranking of layers/heads are very similar. In particular, the relative ordering of heads is largely preserved, so percentile-based selection strategies such as selecting the top $x\%$ most continuous heads are unaffected. This indicates that q-similarity has good stability and generalization across datasets, and that only a small amount of data is needed to calibrate q-similarity for a given model, without requiring separate tuning for each task.

## G    EXPERIMENT DETAILS

### G.1    DETAILS FOR KV CACHE COMPRESSION

**Implementation details.** Following CAKE (Qin et al., 2025), we introduce an adjusted per-layer performance score that incorporates q-similarity derived from TAPPA:

$$P'_l = P_l + \alpha(1 - S_l), \tag{17}$$

where $P_l$ denotes the original layer preference score based on entropy and variance of attention patterns (as defined in Equation (6) of CAKE), $S_l$ is the cosine similarity among queries within a recent window, which instantiates q-similarity in TAPPA, and $\alpha$ is a hyperparameter controlling the contribution of q-similarity. Formally,

$$S_l = \text{sim}(Q_{[-Sw:]}), \tag{18}$$

The intuition is that lower q-similarity indicates a more random and dispersed attention pattern, which generally requires allocating a larger budget. By adjusting $P_l$ with $(1 - S_l)$, we bias the score toward layers exhibiting retrieval-like behaviors.

Finally, following the allocation rule in CAKE, we normalize the adjusted scores to distribute the total budget across layers:

$$B_l = \frac{P'_l}{\sum_{k=0}^{L-1} P'_k} \cdot B_{\text{total}}. \tag{19}$$

**LLMs, benchmark and baselines.** We evaluate our method on Llama-3.1-8B (Dubey et al., 2024) and Qwen2.5-7B (Yang et al., 2024a), using the **LongBench** (Bai et al., 2024) benchmark, which covers 16 long-context understanding tasks.

### G.1.1    BASELINES OF KV CACHE COMPRESSION

Baselines include StreamingLLM (Xiao et al., 2023), H2O (Zhang et al., 2024), SnapKV (Li et al., 2024), PyramidKV (Cai et al., 2025), and CAKE (Qin et al., 2025). We provide detailed descriptions of these baselines in Appendix G.1.1. In the KV cache compression task, we evaluate our method against five representative baselines. Based on whether the budget allocation across layers is uniform, these baselines can be categorized into Uniform Allocation, represented by *StreamingLLM* (Xiao et al., 2023), *H2O* (Zhang et al., 2024), and *SnapKV* (Li et al., 2024), and Non-Uniform Allocation, represented by *PyramidKV* (Cai et al., 2025) and *CAKE* (Qin et al., 2025).

- *StreamingLLM*: retains the first and most recent tokens.
- *H2O*: prioritizes tokens with high cumulative attention.
- *SnapKV*: leverages an observation window at the end of the input to cluster and preserve important KV positions for each head.
- *PyramidKV*: allocates larger budgets to lower layers and smaller ones to higher layers with SnapKV's eviction indicator.
- *CAKE*: introduces a preference-prioritized adaptive allocation strategy, dynamically adjusting budgets across layers.

Table 3: Comparison with additional structured pruning baselines on Llama-2-7B. Results of LLM-Pruner, SliceGPT, LaCo, and ShortGPT are quoted from ShortGPT, and the LLMPruner setting excludes post-training as in ShortGPT.

| Method | Pruning Ratio | PIQA | HellaSwag | WSC | BoolQ | RACE-H | Avg. |
|---|---|---|---|---|---|---|---|
| LLMPruner | 27.00% | 71.22 | 56.46 | 36.54 | 55.20 | 22.56 | 48.40 |
| SliceGPT | 26.40% | 66.21 | 50.27 | 36.54 | 38.32 | 21.07 | 42.48 |
| LaCo | 27.10% | 69.80 | 55.69 | 40.38 | 64.07 | 22.61 | 50.51 |
| ShortGPT | 27.10% | 66.43 | 53.02 | 52.46 | **74.71** | 32.25 | 55.77 |
| TAPPA | **28.10%** | **66.76** | **55.97** | 68.13 | 62.17 | **33.88** | **57.38** |

## G.2 DETAILS FOR LLM PRUNING

**Implementation details.** Building on the Block Influence (BI) metric proposed by ShortGPT (Men et al., 2025), we design an adjusted proxy score:

$$BI' = BI + \beta(1 - q), \tag{20}$$

where $\beta$ is a hyperparameter and $1 - q$ is an importance score derived from q-similarity $q$ in TAPPA.

Following ShortGPT's pruning pipeline, we use the PG19 dataset (Rae et al., 2019) as a calibration set. First, we collect hidden states and queries from each layer while running inference on the calibration data. Next, we compute the proxy scores for all layers based on the adjusted BI score. Finally, we sort the layers in ascending order of scores and remove those with the lowest scores. The number of pruned layers can be adjusted to balance efficiency gains and accuracy preservation.

**LLMs, benchmark, and baselines.** We evaluate our method on Llama-2-7B (Touvron et al., 2023), Llama-3.1-8B (Dubey et al., 2024) and Qwen2.5-7B (Yang et al., 2024a). Using the procedure described above, we first evaluate how redundant each layer is and decide which layers are to be pruned. Then we perform zero-shot task classification on common sense reasoning datasets: PIQA (Bisk et al., 2019), HellaSwag (Zellers et al., 2019), WinoGrande (Sakaguchi et al., 2019) and ARC-easy (Clark et al., 2018) at different pruning ratios. In the main experiments, we compare our method with ShortGPT as the primary baseline. We further include additional structured pruning baselines in Appendix G.2.1. We list the removed layers in Table 4 of Appendix G.2.2.

### G.2.1 COMPARISON WITH ADDITIONAL STRUCTURED PRUNING BASELINES

To provide a more comprehensive comparison for structured LLM pruning, we additionally consider three representative baselines, namely LLMPruner (Ma et al., 2023), SliceGPT (Ashkboos et al., 2024), and LaCo (Yang et al., 2024b). LLMPruner is a gradient-based structured pruning method that removes non-critical coupled structures to preserve model functionality. SliceGPT is a post-training pruning scheme that compresses the model by applying principal component analysis to hidden states in each layer and reducing the embedding dimension. LaCo is a structured pruning approach based on layer collapse, which gradually merges similar layers while using a threshold to avoid excessive collapsing. Following the evaluation protocol in ShortGPT (Men et al., 2025), we report results on PIQA, HellaSwag, WSC, BoolQ, and RACE-H. The results of LLMPruner, SliceGPT, LaCo, and ShortGPT are quoted from ShortGPT, and the LLMPruner setting excludes post-training as in ShortGPT for fair comparison.

As shown in Table 3, our method achieves the best average performance under a higher pruning ratio. In particular, it improves the average score over ShortGPT while maintaining strong accuracy on WSC and RACE-H, indicating that the proposed pruning criterion remains effective in more challenging structured pruning regimes.

### G.2.2 LIST OF REMOVED LAYERS

In the LLM Pruning downstream task, we evaluated our pruning method on different LLMs and pruning ratios. we list the removed layers in Table 4.

Table 4: Removed layers for different benchmark models, using PG19 as calibration dataset.

| Model | Method | Pruning Ratio | Removed Layers |
|---|---|---|---|
| **Llama-2-7B** | ShortGPT | 31% | 21, 22, 23, 24, 25, 26, 27, 28, 29, 30 |
| | TAPPA | 31% | 19, 21, 22, 23, 24, 25, 26, 27, 28, 29 |
| | ShortGPT | 34% | 19, 21, 22, 23, 24, 25, 26, 27, 28, 29, 30 |
| | TAPPA | 34% | 19, 20, 21, 22, 23, 24, 25, 26, 27, 28, 29 |
| **Llama-3.1-8B** | ShortGPT | 28% | 20, 22, 23, 24, 25, 26, 27, 28, 29 |
| | TAPPA | 28% | 21, 22, 23, 24, 25, 26, 27, 28, 29 |
| | ShortGPT | 31% | 20, 21, 22, 23, 24, 25, 26, 27, 28, 29 |
| | TAPPA | 31% | 19, 21, 22, 23, 24, 25, 26, 27, 28, 29 |
| **Qwen2.5-7B** | ShortGPT | 39% | 4, 5, 6, 7, 8, 10, 11, 12, 14, 15, 20 |
| | TAPPA | 39% | 4, 10, 11, 12, 13, 14, 15, 16, 17, 18, 20 |
| | ShortGPT | 43% | 11, 12, 13, 14, 15, 16, 17, 18, 19, 20, 21, 23 |
| | TAPPA | 43% | 4, 5, 10, 11, 12, 13, 14, 15, 16, 17, 18, 20 |

Table 5: Comparison of eight q-similarity formulations on Llama-3.1-8B under the 1024 KV budget on LongBench subsets.

| Method | MF-en | HotpotQA | QMSum | TriviaQA | Lcc | Avg. |
|---|---|---|---|---|---|---|
| q_sim_cosine | 52.63 | 55.45 | 24.59 | **92.04** | 64.58 | **57.86** |
| q_sim_dot | 52.57 | 55.12 | 24.71 | 91.87 | 64.52 | 57.76 |
| q_sim_pearson | **52.68** | 54.84 | 24.80 | 92.03 | **64.91** | 57.85 |
| q_sim_euclidean | 52.58 | 54.85 | 24.76 | 91.62 | 64.59 | 57.68 |
| q_sim_l1 | 52.57 | 54.76 | 24.69 | 91.79 | 64.56 | 57.67 |
| q_sim_angular | 52.59 | 55.05 | **24.87** | 91.86 | 64.82 | 57.84 |
| q_sim_rbf | 52.40 | 54.63 | 24.43 | 91.61 | 64.82 | 57.58 |
| q_sim_kl | 52.54 | **55.56** | 24.56 | 91.48 | 64.54 | 57.74 |

# H ADDITIONAL ABLATIONS AND HYPERPARAMETER SENSITIVITY

## H.1 ALTERNATIVE SIMILARITY FORMULATIONS FOR Q-SIMILARITY

Throughout the paper, we instantiate q-similarity using cosine similarity, which is used to compute the layer-wise score for KV cache compression. To assess whether the downstream gains rely on a particular similarity formulation, we replace cosine similarity with seven alternatives and re-evaluate KV cache compression on representative LongBench subsets. Specifically, we consider dot-product similarity, Pearson correlation coefficient, Euclidean distance, L1 distance, angular distance, radial basis function (RBF) kernel similarity, and Kullback-Leibler (KL) divergence. For distance-based and divergence-based measures, lower values indicate higher similarity and are used accordingly in the layer scoring.

The performance variation across similarity formulations is small, indicating that the method does not critically depend on a particular metric choice. Cosine similarity achieves the best average performance and is therefore used as the default throughout the main experiments.

## H.2 SENSITIVITY TO $\alpha$ IN KV CACHE COMPRESSION

We study the sensitivity to the weighting coefficient $\alpha$ in the adjusted layer score used for KV cache compression. Table 6 reports results on two LongBench subsets under a fixed 1024 KV budget. The performance remains stable across a wide range of $\alpha$ values, and larger $\alpha$ generally yields slightly better results, consistent with the usefulness of the q-similarity signal.

Hyperparameter selection follows a lightweight per-model strategy. For each model, we evaluate $\alpha \in \{0.1, 1, \infty\}$ on a small validation split and select the best $\alpha$. The selected $\alpha$ is then fixed for all

Table 6: Sensitivity to $\alpha$ on Llama-3.1-8B under the 1024 KV budget on LongBench subsets. The setting $\alpha = 0$ corresponds to CAKE without q-similarity. The setting $\alpha = \infty$ corresponds to using only q-similarity.

| $\alpha$ | MF-en | HotpotQA | Avg. |
|---|---|---|---|
| 0 (CAKE) | 52.16 | 55.43 | 53.80 |
| 0.1 | 52.19 | 55.43 | 53.81 |
| 0.2 | 52.17 | 55.39 | 53.78 |
| 0.5 | 52.17 | 55.39 | 53.78 |
| 0.8 | 52.22 | 55.43 | 53.83 |
| 1 | 52.14 | 55.43 | 53.79 |
| 1.5 | 52.19 | 55.42 | 53.80 |
| 2 | 52.27 | 55.42 | 53.84 |
| 5 | 52.11 | **55.56** | 53.84 |
| 10 | 52.24 | 55.54 | 53.89 |
| $\infty$ | **52.41** | 55.44 | **53.93** |

Table 7: Per-layer computational overhead comparison between q-similarity computation and CAKE under different context lengths. Latency is measured in ms, and memory is measured in MB.

| Length | CAKE | | q-sim | | Improvement | |
|---|---|---|---|---|---|---|
| | Latency | Memory | Latency | Memory | Latency | Memory |
| 4K | 0.198 | 72.12 | 0.195 | 8.69 | 2% | 88% |
| 8K | 0.308 | 136.12 | 0.197 | 8.69 | 36% | 94% |
| 16K | 0.506 | 264.12 | 0.199 | 8.69 | 61% | 97% |
| 32K | 0.874 | 520.12 | 0.196 | 8.69 | **78%** | **98%** |

datasets and KV budgets for that model in the full evaluation. The setting $\alpha = \infty$ corresponds to using only the q-similarity term to rank layers.

### H.3 COMPUTATIONAL OVERHEAD OF Q-SIMILARITY COMPUTATION

This subsection evaluates the runtime and memory overhead of computing q-similarity scores during inference. We measure the per-layer overhead of our q-similarity computation and compare it with CAKE, which maintains attention score-based statistics for eviction. All measurements are conducted on Llama-3.1-8B with a window size of 32, under different context lengths up to 32K tokens.

As shown in Table 7, the overhead of q-similarity is small and stable across all context lengths. In particular, the per-layer latency stays below 0.2 ms, and the additional memory consumption remains about 8.69 MB. This behavior is expected because q-similarity only computes query similarities within a fixed window, so its complexity is effectively independent of the total sequence length. In contrast, attention score-based methods like CAKE update and store per-token statistics to derive eviction signals, so both runtime and memory overhead increase with the context length. At 32K tokens, q-similarity reduces the per-layer latency overhead by **78%** and the additional memory consumption by **98%** compared to CAKE, highlighting the efficiency advantage of a query-based signal for budget allocation.

### H.4 SENSITIVITY TO $\beta$ IN LAYER PRUNING

We conduct a sensitivity study for the weighting coefficient $\beta$ used in layer pruning. As shown in Table 8, the performance varies moderately across $\beta$, with a plateau for $\beta \in [0.2, 0.4]$. Identical results for nearby $\beta$ values occur when the induced pruned layer sets are unchanged.

Table 8: Sensitivity to $\beta$ on Llama-2-7B under a 34% pruning ratio. Identical results for some $\beta$ values arise when the corresponding pruned layer sets are the same.

| $\beta$ | Piqa | Hellaswag | Winogrande | Arc Easy | Average |
|---|---|---|---|---|---|
| 0 | 60.83 | 42.11 | 60.38 | **44.15** | 51.87 |
| 0.1 | 60.83 | 42.11 | 60.38 | **44.15** | 51.87 |
| 0.2 | 60.45 | **48.53** | **62.43** | 42.55 | **53.49** |
| 0.3 | 60.45 | **48.53** | **62.43** | 42.55 | **53.49** |
| 0.4 | 60.45 | **48.53** | **62.43** | 42.55 | **53.49** |
| 0.5 | **61.10** | 46.47 | 57.77 | 39.69 | 51.26 |

Hyperparameter selection follows a per-model strategy. For each model, we evaluate $\beta \in \{0.1, 0.3, 0.5\}$ on a small validation split and select the best $\beta$. The selected $\beta$ is then fixed for all datasets and pruning ratios for that model in the full evaluation.

# I    COMPARISON WITH DUOATTENTION

In this section, we provide a detailed comparison between the q-similarity based method derived from TAPPA and DuoAttention (Xiao et al., 2024), a recent baseline that explicitly distinguishes retrieval heads and streaming heads for KV cache compression.

## I.1    BASELINES AND METHODOLOGY ADAPTATION

**DuoAttention** is an optimization-based method that explicitly identifies retrieval heads via training. It assigns a learnable scalar, which we denote as $\alpha_{\mathrm{duo}}$, to each attention head to represent its retrieval importance.

To conduct a direct comparison between the q-similarity metric derived from TAPPA and DuoAttention's learned importance for the layer-wise budget allocation task, we adapted their scoring mechanism into the same layer-wise budget allocation scheme. Specifically, we calculate the importance score for each layer $l$ by averaging the $\alpha_{\mathrm{duo}}$ values across all heads in that layer. We then compute the allocated budget $B_l$ for layer $l$ using a formulation analogous to Eq. 19:

$$B_l = \frac{\bar{\alpha}_{\mathrm{duo}}^{(l)}}{\sum_{k=0}^{L-1} \bar{\alpha}_{\mathrm{duo}}^{(k)}} \cdot B_{\mathrm{total}}, \tag{21}$$

where $\bar{\alpha}_{\mathrm{duo}}^{(l)}$ is the average score of layer $l$. This setup allows us to fairly evaluate the effectiveness of the two metrics in identifying layers that require higher KV cache budgets.

Table 9: Performance comparison with DuoAttention on LongBench using Llama-3.1-8B. For each budget, the best score on each subset is highlighted in bold.

| Budget | Method | Single-DocumentQA | | | Multi-DocumentQA | | | Summary | | | Few-shot Learning | | | Synthetic | | Code | | Average↑ |
|---|---|---|---|---|---|---|---|---|---|---|---|---|---|---|---|---|---|---|
| | | NrtvQA | Qasper | MF-en | HotpotQA | 2WikiMQA | Musique | GovReport | QMSum | MultiNews | TREC | TriviaQA | SAMSum | PCount | PRe | Lcc | RB-P | |
| Full | Full | 31.06 | 45.43 | 53.78 | 55.04 | 47.14 | 31.29 | 34.87 | 25.33 | 27.49 | 72.50 | 91.25 | 43.81 | 6.00 | 99.50 | 63.36 | 56.65 | 49.06 |
| 512 | DuoAttention | **30.60** | **43.09** | **51.70** | 54.15 | 45.98 | 30.51 | **25.62** | 24.50 | **24.88** | 62.50 | 92.35 | 41.70 | 6.08 | 99.50 | 64.55 | 56.87 | 47.16 |
| | TAPPA | 29.47 | 42.66 | 51.63 | **54.53** | **46.64** | **30.81** | 25.48 | **24.57** | 24.71 | 62.50 | 92.35 | 42.42 | 6.25 | 99.50 | 64.56 | 57.35 | **47.21** |
| 1024 | DuoAttention | 30.13 | 44.57 | **52.72** | 54.58 | 46.95 | 31.01 | 28.16 | 24.62 | 26.22 | 67.00 | 91.89 | 42.22 | 6.50 | 99.50 | 64.78 | 58.84 | 48.11 |
| | TAPPA | **30.77** | **44.94** | 52.14 | **55.43** | **46.99** | **31.16** | **28.72** | **24.90** | **26.65** | 69.50 | 91.95 | 42.38 | 6.00 | 99.50 | 64.99 | 58.84 | **48.43** |
| 2048 | DuoAttention | 30.63 | 45.39 | **53.62** | 55.49 | 46.52 | 30.32 | **30.61** | **24.98** | **27.25** | 70.50 | 91.49 | 42.74 | 6.00 | 99.50 | 64.96 | 58.82 | 48.68 |
| | TAPPA | **30.70** | **45.69** | 53.06 | 55.49 | **46.68** | **30.94** | 30.54 | 24.65 | 27.12 | 71.00 | 91.65 | 43.00 | 6.00 | 99.50 | 64.93 | 58.80 | **48.73** |

## I.2    POTENTIAL FOR HIGHER COMPRESSION RATIO

It is crucial to highlight the fundamental difference in how the two methods categorize attention patterns and the resulting impact on the compression scope. DuoAttention operates on a binary premise where it differentiates *Streaming Heads* that necessitate only sink and recent tokens from

Table 10: Performance comparison of Expected Attention (EA) with and without our q-similarity based budget allocation on LongBench. For each budget, the best score on each subset is highlighted in bold.

| Budget | Method | Single-DocumentQA | | | Multi-DocumentQA | | | Summary | | | Few-shot Learning | | | Synthetic | | Code | | Average↑ |
|---|---|---|---|---|---|---|---|---|---|---|---|---|---|---|---|---|---|---|
| | | NrtvQA | Qasper | MF-en | HotpotQA | 2WikiMQA | Musique | GovReport | QMSum | MultiNews | TREC | TriviaQA | SAMSum | PCount | PRe | Lcc | RB-P | |
| | | | | | | | | **Llama-3.1-8B** | | | | | | | | | | |
| 512 | EA | 21.53 | **37.72** | 40.01 | **47.59** | **40.99** | 17.09 | 27.77 | **22.99** | 26.43 | **47.50** | **72.82** | 35.60 | 7.17 | 81.50 | 52.80 | **50.39** | 39.37 |
| | EA with TAPPA | **21.72** | 34.46 | **41.15** | 46.75 | 40.92 | **19.50** | **27.99** | 22.62 | **26.68** | 45.00 | 71.83 | **36.94** | **7.85** | **84.50** | **54.18** | 49.32 | **39.46** |
| 1024 | EA | 23.36 | **40.79** | 42.72 | **52.30** | 44.53 | 18.19 | **29.66** | **23.90** | 27.17 | **55.50** | **88.70** | 36.23 | 4.79 | 88.00 | 52.97 | 53.42 | 42.64 |
| | EA with TAPPA | **29.17** | 38.58 | **43.13** | 50.30 | **45.91** | **23.78** | 29.37 | 23.85 | **27.21** | 53.50 | 87.30 | **36.86** | **4.88** | **90.50** | **55.78** | **54.25** | **43.40** |
| 2048 | EA | 25.68 | 42.58 | 47.13 | 49.91 | 43.40 | 18.88 | **31.70** | 23.69 | **27.37** | **62.00** | 86.25 | 36.96 | **7.25** | **95.50** | 52.00 | 51.67 | 43.87 |
| | EA with TAPPA | **30.13** | **43.96** | **50.20** | **50.57** | **44.54** | **23.26** | 31.25 | **23.96** | 27.35 | 58.50 | **86.71** | **38.22** | 6.92 | 95.00 | **54.16** | **54.52** | **44.95** |
| | | | | | | | | **Qwen2.5-7B** | | | | | | | | | | |
| 512 | EA | **19.30** | 24.62 | 27.85 | 24.63 | 23.73 | 12.20 | 26.81 | **21.27** | 23.70 | 14.00 | 71.42 | 35.29 | 4.17 | 7.50 | 19.68 | 31.79 | 24.25 |
| | EA with TAPPA | 17.40 | **31.09** | **34.51** | **26.86** | **31.39** | **14.90** | **28.41** | 20.76 | **24.39** | **30.75** | **71.87** | **43.00** | **6.03** | **89.08** | **54.49** | **44.46** | **35.59** |
| 1024 | EA | 18.57 | **35.71** | 38.93 | 36.81 | **38.43** | **18.70** | **30.07** | 20.68 | 25.07 | 31.50 | **82.48** | 37.06 | **6.25** | 94.33 | 32.66 | 37.93 | 36.57 |
| | EA with TAPPA | **21.39** | 35.21 | **42.03** | **36.92** | 36.31 | 17.68 | 29.87 | **21.40** | **25.09** | **45.50** | 76.54 | **43.81** | 5.12 | **97.00** | **52.53** | **52.21** | **39.91** |
| 2048 | EA | 20.28 | **41.08** | 44.97 | **45.46** | **39.47** | **25.36** | 31.45 | 21.64 | **25.66** | **55.05** | **86.91** | 38.38 | **7.65** | **98.25** | 34.76 | 41.55 | 41.12 |
| | EA with TAPPA | **23.47** | 39.95 | **46.55** | 45.38 | 38.35 | 25.11 | **31.47** | **21.93** | 25.42 | 55.00 | 82.59 | **44.39** | 6.08 | 97.33 | **41.84** | **55.01** | **42.49** |

*Retrieval Heads* requiring full history retention. Consequently, its compression efforts primarily focus on heads exhibiting streaming behavior.

In contrast, TAPPA provides a more detailed categorization. The q-similarity metric derived from TAPPA distinguishes complex *Retrieval* patterns from a variety of regular attention patterns, including Re-access, Sequential, and Seasonal patterns. Crucially, TAPPA identifies these regular patterns as compressible. This effectively expands the scope of compressible heads beyond just streaming heads. By compressing these additional heads that might otherwise be preserved, this strategy could achieve a higher compression ratio while maintaining model performance.

### I.3 EXPERIMENTAL SETUP

We compare DuoAttention and TAPPA under strict KV cache budgets. For DuoAttention, the coefficient $\alpha_{\text{duo}}$ is set to the official value trained on Llama-3.1-8B and released by the authors.[1] All results are evaluated on the LongBench benchmark with 16 subsets. We report performance at KV cache budgets of 512, 1024, and 2048 tokens, and also include the full context baseline.

### I.4 RESULTS AND ANALYSIS

The results are in Table 9. Across all three budgets, the q-similarity based method achieves higher average performance than the DuoAttention based allocation. At the subset level, the advantage is most visible on multihop reasoning and retrieval-oriented tasks. For example, at the 1024 budget, TAPPA improves HotpotQA from 54.58 to 55.43 and TREC from 67.00 to 69.50. At the 2048 budget, the q-similarity based method reaches an average score of 48.73, which is close to the full context baseline of 49.06.

## J COMPARISON WITH EXPECTED ATTENTION

Expected Attention (Devoto et al., 2025) is a training-free KV cache compression method that ranks and prunes KV pairs by analytically estimating how future queries are expected to attend to cached keys. Expected Attention uses a uniform layerwise budget allocation and then applies its expected attention-based importance score within each layer to perform KV pruning. To evaluate the compatibility of the budget adjustment strategy derived from TAPPA with other compression frameworks, we integrate the q-similarity based layerwise budget allocation into Expected Attention by replacing the uniform allocation while keeping the original Expected Attention scoring and pruning mechanism unchanged. We use the official hyperparameter settings of Expected Attention from the released implementation and do not tune them for our integration.[2]

---

[1] https://github.com/mit-han-lab/duo-attention.git
[2] https://github.com/NVIDIA/kvpress

Table 10 reports results on LongBench with 16 subsets under KV budgets of 512, 1024, and 2048. Across both backbones and all budgets, adding TAPPA budget allocation improves the average performance over Expected Attention. Specifically, the improvement is approximately **46.8%** on Qwen-2.5 with the 512 KV budget, increasing the average score from 24.25 to 35.59. These results indicate that the proposed q-similarity based temporal signal can serve as a plug-in budget allocation component that strengthens Expected Attention without modifying its core expected attention estimation.

## K  BROADER RELEVANCE OF ATTENTION PATTERN ANALYSIS.

Attention pattern analysis benefits both training and inference of large language models. During inference, long reasoning Wang et al. (2025) operates over long contexts, yet only a small subset of tokens and attention routes are informative for each step. Distinguishing stable allocation from retrieval-oriented allocation can prioritize evidence tokens and reduce redundant computation while preserving reasoning-relevant information flow. For training, RL training Liang et al. (2025; 2026); Hao et al. (2025) is increasingly influential but remains costly due to long horizon rollouts and repeated forward passes; pattern-based diagnostics can guide architectures and objectives that improve information density and yield models that more readily admit sparsification or quantization. Attention patterns also support model editing Ma et al. (2025) by localizing the attention routes that mediate specific behaviors, enabling targeted interventions with fewer unintended side effects.

## L  THE USE OF LARGE LANGUAGE MODELS (LLMS)

Large Language Models (LLMs) were employed solely for the purpose of enhancing the linguistic clarity and stylistic refinement of this manuscript.

