# OpenReview forum: "Why Attention Patterns Exist: A Unifying Temporal Perspective Analysis"
_ICLR.cc/2026/Conference — ICLR 2026 Poster_

### Official Review · Reviewer_Qj6A · 2025-10-22

**Soundness:** 3
**Presentation:** 3
**Contribution:** 2
**Rating:** 4
**Confidence:** 4

**Summary:**

This paper proposes a unifying temporal perspective to explain the emergence of diverse attention patterns in LLMs. The authors categorize patterns into predictable (re-access, sequential, seasonal) and unpredictable (retrieval-like) types, attributing the distinction to variations in query self-similarity over time. They provide theoretical analyses linking pattern formation to query-key continuity and RoPE, and validate their framework through applications in KV cache compression and LLM pruning. Experiments on models like Llama and Qwen show consistent improvements over baseline methods.

**Strengths:**

- The paper is well-organized, with clear motivations, methodical explanations, and thorough appendices.
- The paper offers a coherent perspective that integrates previously fragmented observations (e.g., sink heads, diagonal patterns, retrieval heads) under a single temporal continuity lens.
- The paper provides detailed mathematical proofs for each pattern type, clearly linking RoPE mechanics and query-key dynamics to observable attention structures.
- The proposed q-similarity metric is effectively applied to downstream tasks (KV cache compression, pruning), demonstrating improved performance over strong baselines.

**Weaknesses:**

- The entire framework hinges on the assumption of temporal continuity in queries and keys. While this is likely a reasonable assumption for many layers and tasks, its universality is not thoroughly explored. The analysis might be less applicable in layers or for inputs where representations change abruptly. A discussion of the boundaries of this assumption would strengthen the work.
- The q-similarity metric is central to the applications, but its specific formulation (e.g., the choice of cosine similarity) lacks a comprehensive ablation study. It remains unclear how sensitive the performance gains are to these choices, or if an even more effective metric derived from the same theory could be designed.
- The proofs provided in the appendix, while a valuable effort, contain significant weaknesses that undermine their theoretical rigor.

**Questions:**

1.How does q-similarity vary across different layers and heads? Is it consistent across models, or does it require per-model calibration?
2.The paper claims that high q-similarity implies redundancy in pruning. Is this always true? Could some stable patterns be critical for certain tasks (e.g., syntax parsing)?
3.In the proof of vertical stability (Theorem 5.1, Appendix B), a crucial step bounds the change in the angle between the query and a fixed key:
$$|\phi_{t+1,i}^{(m)}-\phi_{t,i}^{(m)}|\leq\frac{\|q_{t+1}^{(m)}-q_t^{(m)}\|}{r_m}$$
However, consider a simplified scenario in 2D: let q_{t}=(1,0) and q_{t+1}=(cos(2arcsin(ε/2)), sin(2arcsin(ε/2))). Here ||q_{t+1}-q_{t}|| ≤ ε， the angle change is 2arcsin(ε/2) ＞ ε. This suggests the above inequality does not hold.

In the proof of Theorem 5.4 (Seasonal Pattern), the derivation for non-dominant channels contains a critical error. The term (i-t)θ_m is incorrectly repeated in both cosine functions when calculating |a_{t+L,i}^{(m)} - a_{t,i}^{(m)}|. The correct expression for a_{t+L,i}^{(m)} should have a phase of (i-t)θ_m - Lθ_m. More critically, the standard inequality used to bound the difference for non-dominant channels is misapplied. This inequality, | ||u|| ||v|| cos φ - ||u'|| ||v'|| cos φ' | ≤ ..., is valid only when φ and φ' are the geometric angles between the vector pairs (u,v) and (u',v'), respectively. In your proof, you apply it with angles φ = φ_{t,i}^{(m)} + (i-t)θ_m and φ' = φ_{t+L,i}^{(m)} + (i-t)θ_m. However, these are not pure geometric angles but include an additive positional phase. This misapplication renders the subsequent bound on non-dominant channels invalid.

---

> ### Author Response · Authors · 2025-11-25
> **Response to Reviewer Qj6A**
>
> Dear Reviewer Qj6A,
>
> Thank you for your **valuable review** and for your **recognition of the clear organization of our presentation, the unifying temporal-continuity perspective on attention patterns, the detailed theoretical analysis, and the effective application**. Your positive evaluation is truly encouraging and means a great deal to us. Below, we provide detailed responses to your insightful and constructive comments. We have incorporated the corresponding discussions and experiments into the appendix and added the necessary references in the main text.
>
> Throughout the discussion phase, we genuinely look forward to your feedback and are fully committed to addressing any remaining concerns. If our responses have properly resolved your questions,  **we would be deeply grateful if you would consider raising your score** . If anything remains unclear or unconvincing,  **we would truly appreciate your further guidance** , and we stand ready to engage promptly and constructively.
>
> > **Q1:** How does q-similarity vary across different layers and heads? Is it consistent across models, or does it require per-model calibration?
>
> **A:** To better understand the behavior of q-similarity, we compute per-head q-similarity scores across all layers for two models (Llama-3.1 and Qwen-2.5) on two representative datasets (GSM8K and AIGC). We summarize the main observations below, and include the full visualizations in the revised  **Appendix F.3** .
>
> 1. **Overall high q-similarity, supporting the temporal continuity assumption.**
>    Across all heads and layers, the average q-similarity is high for both models (around 0.80 for Llama-3.1 and 0.86 for Qwen-2.5). This empirically supports our working assumption that queries tend to evolve in a temporally continuous manner in a large portion of the network.
> 2. **Model-specific but layer-structured distributions.**
>    Each model exhibits its own characteristic distribution of q-similarity values, indicating that q-similarity reflects model-specific properties and thus naturally calls for per-model calibration. At the same time, within a given model we observe a clear and consistent structure: heads in the *same* layer have similar q-similarity scores (forming tight clusters), whereas the average q-similarity differs significantly *across* layers. This justifies our design choice of operating at the layer level (e.g., using layer-wise averages) when building downstream metrics and policies.
> 3. **Stable across datasets for the same model, enabling lightweight calibration.**
>    For a fixed model, the q-similarity distribution is highly consistent across datasets. For Llama-3.1, the average q-similarity on GSM8K and AIGC differs by only about 0.01. For Qwen-2.5, the absolute mean difference between the two datasets is about 0.07, but the overall shape and ranking of layers/heads are very similar. In particular, the relative ordering of heads is largely preserved, so percentile-based selection strategies (e.g., “top x% most unpredictable heads”) are unaffected. This indicates that q-similarity has good stability and generalization across datasets, and that only a small amount of data is needed to calibrate q-similarity for a given model, without requiring separate tuning for each task.

---

> ### Author Response · Authors · 2025-11-25
> **Response to Reviewer Qj6A**
>
> > **W1:** The entire framework hinges on the assumption of temporal continuity in queries and keys. While this is likely a reasonable assumption for many layers and tasks, its universality is not thoroughly explored. The analysis might be less applicable in layers or for inputs where representations change abruptly. A discussion of the boundaries of this assumption would strengthen the work.
>
> **A:** Thank you for raising this important point. We fully agree that temporal continuity of queries and keys is not a universal property of all layers and all inputs. Our framework is explicitly designed to distinguish between cases where this assumption holds (predictable patterns) and cases where it does not (unpredictable, retrieval-like patterns).
>
> 1. **Conceptual scope: predictable vs. unpredictable cases.** In Section 4, we already classify attention patterns into *predictable* and *unpredictable* categories from a time-series perspective. Predictable patterns are defined as those where the high-attention indices evolve smoothly over time, while unpredictable patterns exhibit irregular jumps with little temporal consistency. In our framework, **low q-similarity is the signal that a head belongs to unpredictable case.** For such heads and layers, we *do not* apply the fine-grained theoretical characterizations of re-access, sequential, or seasonal patterns; instead, **we treat them as retrieval-like and allocate larger KV budgets** in our applications. We will make this design choice explicit at the end of Section 4 to clearly delineate where the continuity-based analysis is intended to apply.
> 2. **Empirical prevalence and practical scope of continuity.** As detailed in our response to Q1 and in the new  **Appendix F.3** , we systematically measure q-similarity across layers, heads, models, and datasets. The head-wise heatmaps show that: (i) q-similarity is on average high, supporting the presence of temporal continuity in a large portion of the network; (ii) its variation is primarily across layers, with heads in the same layer displaying very similar q-similarity; and (iii) for a fixed model, the q-similarity distribution is highly consistent across datasets. These observations indicate that a substantial fraction of layers operate in a high-continuity case where our theoretical analysis is directly applicable, while still leaving room for low-continuity, retrieval-like heads that are treated conservatively.
> 3. **Clarifying the boundaries in the revised manuscript.** In the revision, we have (i) add a short paragraph at the end of Section 4 explicitly stating that our detailed theoretical results are meant for heads with high temporal continuity; and (ii) refer to the new Appendix F.3 analysis to empirically illustrate how often the continuity assumption holds across different layers, models, and datasets. We hope this clearer delineation of scope and the additional empirical evidence address the reviewer’s concern about the universality and boundaries of the continuity assumption.
>
> > **W2:** The q-similarity metric is central to the applications, but its specific formulation (e.g., the choice of cosine similarity) lacks a comprehensive ablation study. It remains unclear how sensitive the performance gains are to these choices, or if an even more effective metric derived from the same theory could be designed.
>
> **A:** In response to the reviewer's concern regarding the lack of comprehensive ablation studies on the formulation of  *q-similarity* , we have added experiments comparing different similarity metrics. Specifically, we evaluate eight variants—cosine similarity, dot-product similarity, Pearson correlation coefficient, Euclidean distance, L1 distance, angular distance, radial basis function (RBF) kernel similarity, and KL divergence—on five representative LongBench subsets.
>
> As shown in Table 1, the choice of metric has only a limited impact on the final performance, and the overall variation across metrics is small. This indicates that our *q-similarity* measure is robust and does not rely on any particular similarity formulation. Among all metrics, cosine similarity achieves the best average performance, providing ablation support for its use in our main experiments.
>
> ***Table 1** : Comparison of eight q-similarity formulations on Llama-3.1-8B under 1024 budget (Longbench).*
>
> |   Method   | MF-en | HotpotQA | QMSum | TriviaQA |  Lcc  |    Avg.    |
> | :----: | :---: | :---: | :---: | :------: | :---: | :---: |
> |  cosine  | 52.63 |  55.45  | 24.59 |  92.04  | 64.58 | **57.86** |
> | dot | 52.57 |  55.12  | 24.71 |  91.87  | 64.52 |57.76|
> | pearson | 52.68 |  54.84  | 24.8 |  92.03  | 64.91 |  57.85 |
> | euclidean | 52.58 |  54.85  | 24.76 |  91.62  | 64.59 | 57.68 |
> | l1 | 52.57 |  54.76  | 24.69 |  91.79  | 64.56 |57.67 |
> | angular| 52.59 |  55.05  | 24.87 |  91.86  | 64.82 | 57.84 |
> |rbf| 52.4 |  54.63  | 24.43 |  91.61  | 64.82 | 57.58  |
> | kl | 52.54 |  55.56  | 24.56 |  91.48  | 64.54 |57.74 |

---

> ### Author Response · Authors · 2025-11-25
> **Response to Reviewer Qj6A**
>
> > **Q2:** The paper claims that high q-similarity implies redundancy in pruning. Is this always true? Could some stable patterns be critical for certain tasks (e.g., syntax parsing)?
>
> **A:** We thank the reviewer for the question regarding the relationship between high Q-similarity and pruning redundancy. Our response is as follows.
>
> 1. **Regarding the definition of redundancy in layer pruning**. As established in prior works (e.g., ShortGPT), the fundamental principle of layer pruning is to identify "redundant layers." A layer is considered redundant if its input and output hidden states are highly similar (measured by Block Influence), indicating that the layer contributes minimal transformation or new information to the hidden states. Such layers can be safely pruned without significantly affecting model performance.
> 2. **Why high Q-similarity can indicate redundancy.** We propose Q-similarity as a critical indicator of layer importance based on the distinct roles of attention heads:
>
>    - **Low Q-similarity (Retrieval Heads):** As demonstrated in our paper, low Q-similarity correlates with  *retrieval patterns* . Recent studies (e.g., KVTuner [1], DuoAttention [2]) emphasize that **retrieval heads are essential for handling complex tasks** and long-context understanding. Therefore, layers containing these dynamic, low-similarity heads are critical and should be preserved.
>    - **High Q-similarity (Stable Patterns):** Conversely, high Q-similarity points to stable, repetitive patterns like attention sinks. These are common and redundant across the model. Thus, **pruning layers with stable patterns is safer** and impacts performance much less than losing unique retrieval heads.
>
>    Consequently, layers with high Q-similarity are less likely to possess the critical retrieval capabilities required for performance. By combining Block Influence with Q-similarity, we can more accurately identify and prune layers that are both transformationally inactive and functionally less critical.
> 3. **On stable, high-similarity patterns that remain functionally important.** Weagree with the reviewer that some stable patterns are fundamental to model capabilities. However, our strategy does not remove all layers with stable patterns; instead, it selects layers based on  **relative redundancy across layers** . Even among stable patterns, different layers exhibit varying degrees of stability. We prune only those layers with the highest Q-similarity (most stable) and lowest Block Influence (least transformational contribution). In practice, layers carrying essential structural or foundational capabilities typically do not rank highest under this combined criterion and are thus preserved.
>
> Our experimental results (Table 2 in our paper) demonstrate that this approach preserves the model's foundational capabilities on structured tasks while achieving consistent performance improvements, validating the effectiveness of this pruning strategy.
>
> [1] Li, Xing, et al. "KVTuner: Sensitivity-Aware Layer-Wise Mixed-Precision KV Cache Quantization for Efficient and Nearly Lossless LLM Inference." ICML 2025.
>
> [2] Xiao, Guangxuan, et al. "Duoattention: Efficient long-context llm inference with retrieval and streaming heads." ICLR 2025.

---

> ### Author Response · Authors · 2025-11-25
> **Response to Reviewer Qj6A**
>
> > **W3 & Q3**: The proofs provided in the appendix, while a valuable effort, contain significant weaknesses that undermine their theoretical rigor.
>
> > **(1)** In the proof of vertical stability (Theorem 5.1, Appendix B), a crucial step bounds the change in the angle between the query and a fixed key:However, consider a simplified scenario in 2D: let $q_{t}=(1,0)$ and $q_{t+1}=(cos(2arcsin(ε/2)), sin(2arcsin(ε/2)))$. Here $||q_{t+1}-q_{t}|| ≤ ε$， the angle change is 2arcsin(ε/2) ＞ ε. This suggests the above inequality does not hold.
>
> **Regarding the issues with** **the proof of Theorem 5.1 in Appendix B (Vertical Stability)** :
>
> We thank the reviewer for the astute observation regarding the geometric inequality in Appendix B. You are correct that strictly speaking, the arc length is always greater than the chord length. Specifically, for a radius $r_m$ and angular change $\Delta \phi$, the chord length is
>
> $$
> \varepsilon = 2r_m \sin\left(\frac{\Delta \phi}{2}\right),
> $$
>
> implying
>
> $$
> \Delta \phi = 2 \arcsin\left(\frac{\varepsilon}{2r_m}\right).
> $$
>
> Thus, our original statement $|\Delta \phi| \leq \varepsilon / r_m$ was directionally incorrect.
>
> However, this technical oversight does not affect the validity or the asymptotic conclusion of Theorem 5.1. We can correct the proof by introducing a geometric constant factor $\frac{\pi}{2}$.
>
> Since the function $f(x) = \frac{\arcsin(x)}{x}$ is monotonically increasing on $(0, 1]$ and bounded by $\frac{\pi}{2}$, we have the inequality
>
> $$
> \arcsin(x) \leq \frac{\pi}{2} x \quad \text{for} \quad x \in [0, 1].
> $$
>
> Applying this to the RoPE channel $m$:
>
> 1. Let $\varepsilon^{(m)} = \|q_{t+1}^{(m)} - q_t^{(m)}\|$ be the Euclidean distance (chord) in the $m$-th 2D subspace.
> 2. The angular change is
>    $$
>    |\phi_{t+1,i}^{(m)} - \phi_{t,i}^{(m)}| = 2 \arcsin\left(\frac{\varepsilon^{(m)}}{2r_m}\right).
>    $$
> 3. Applying the bound:
>    $$
>    |\phi_{t+1,i}^{(m)} - \phi_{t,i}^{(m)}| \leq 2 \cdot \frac{\pi}{2} \cdot \frac{\varepsilon^{(m)}}{2r_m} = \frac{\pi}{2} \frac{\varepsilon^{(m)}}{r_m}.
>    $$
> 4. Since the subspace distance is bounded by the total vector distance ($\varepsilon^{(m)} \leq \|q_{t+1} - q_t\| = \varepsilon$), we arrive at the corrected bound:
>    $$
>    |\phi_{t+1,i}^{(m)} - \phi_{t,i}^{(m)}| \leq \frac{\pi}{2} \frac{\varepsilon}{r_m}.
>    $$
>
> **Impact on Theorem:**
> This correction introduces a constant factor of $\frac{\pi}{2}$ into the first term of Equation 10 and the definition of $\delta$ in Equation 11. The revised bound becomes:
>
> $$
> \delta := \varepsilon \sum_{m=1}^{M} \|k_i^{(m)}\| + \frac{\pi}{2}\varepsilon \sum_{m=1}^{M} \frac{w_m}{r_m} + \sum_{m=1}^{M} w_m |\theta_m|.
> $$
>
> Crucially, as $\varepsilon \to 0$, the terms dependent on $\varepsilon$ still vanish linearly. Thus, **the vertical stability conclusion**—that $\Delta_{t,i}$ becomes uniformly small—**remains entirely valid**. We have update the proof in the revision to reflect this precise geometric bound.

---

> > ### Comment · Reviewer_Qj6A · 2025-11-26
> >
> > The equation $$ |\phi_{t+1,i}^{(m)} - \phi_{t,i}^{(m)}| = 2 \arcsin\left(\frac{\varepsilon^{(m)}}{2r_m}\right). $$ relies on the assumption that $$ \|q_{t+1}^{(m)}\| = \|q_t^{(m)}\| $$. However, this assumption is not necessarily true for LLMs such as Llama.

---

> ### Author Response · Authors · 2025-11-25
> **Response to Reviewer Qj6A**
>
> > **W3&Q3: (2)** In the proof of Theorem 5.4 (Seasonal Pattern), the derivation for non-dominant channels contains a critical error. The term (i-t)θ_m is incorrectly repeated in both cosine functions when calculating |a_{t+L,i}^{(m)} - a_{t,i}^{(m)}|. The correct expression for a_{t+L,i}^{(m)} should have a phase of (i-t)θ_m - Lθ_m. More critically, the standard inequality used to bound the difference for non-dominant channels is misapplied. This inequality, | ||u|| ||v|| cos φ - ||u'|| ||v'|| cos φ' | ≤ ..., is valid only when φ and φ' are the geometric angles between the vector pairs (u,v) and (u',v'), respectively. In your proof, you apply it with angles φ = φ_{t,i}^{(m)} + (i-t)θ_m and φ' = φ_{t+L,i}^{(m)} + (i-t)θ_m. However, these are not pure geometric angles but include an additive positional phase. This misapplication renders the subsequent bound on non-dominant channels invalid.
>
> **Response to the concern on Theorem 5.4 (Seasonal Pattern).** Thank you for the careful reading and for pointing out the issues in the original proof.
>
> 1. **Correction of the RoPE phase term.** You are absolutely right. In the revision, we have removed this incorrect line and **rewritten the derivation in Appendix E (Proof of Theorem 5.4)** by treating RoPE explicitly as an orthogonal rotation matrix on each 2D channel. This avoids any ad-hoc phase expressions and makes the dependence on the offset $L$ precise.
> 2. **Clarification of the inequality and geometric angles.** We agree that, in the previous version, the inequality $|\|u\|\|v\|\cos\phi - \|u'\|\|v'\|\cos\phi'|\le\cdots$ was informally applied to angles of the form $\phi_{t,i}^{(m)} + (i-t)\theta_m$, which mix the geometric angle between $q_t^{(m)}$ and $k_i^{(m)}$ with an additional positional phase. In the revised proof, we no longer work at the “angle + positional phase” level. Instead, for each channel $m$ we define the post-RoPE vectors $\tilde q_t^{(m)} := R_t^{(m)} q_t^{(m)}$ and $\tilde k_i^{(m)} := R_i^{(m)} k_i^{(m)}$, where $R_t^{(m)}$ and $R_i^{(m)}$ are the RoPE rotation matrices. The channel contribution is then written as a dot product $a_{t,i}^{(m)} = \langle \tilde q_t^{(m)}, \tilde k_i^{(m)}\rangle$, and we apply the standard inequality
>
>    $$
>    |u^\top v - u'^\top v'| \le \|v\|\|u-u'\| + \|u'\|\|v-v'\|
>    $$
>
>    directly to these post-RoPE vectors. This removes the need to interpret $\phi_{t,i}^{(m)} + (i-t)\theta_m$ and $\phi_{t+L,i}^{(m)} + (i-t)\theta_m$ as geometric angles and resolves the issue you raised.
> 3. **Resulting bound and main intuition unchanged.** The revised Appendix E now shows that, under (i) the near-resonance condition $|L\theta_{m^\star} - 2k\pi| \le \delta$ for the dominant RoPE channel $m^\star$, and (ii) the $L$-periodicity assumptions on queries and keys, the post-RoPE vectors $\tilde q_t^{(m^\star)}$ and $\tilde q_{t+L}^{(m^\star)}$ (and similarly along the key dimension) remain close, with deviations bounded by $\varepsilon_q,\varepsilon_k$ and $\delta$. Non-dominant channels are uniformly bounded and absorbed into the constants. As a result, the corrected proof preserves the original statement of Theorem 5.4: the dominant RoPE channel induces an approximately seasonal attention pattern with period $L$, while the remaining channels contribute only small perturbations.

---

> > ### Comment · Reviewer_Qj6A · 2025-11-26
> >
> > The author answered all my questions and I will raise my score.

---

> ### Author Response · Authors · 2025-11-27
> **Further Response to Reviewer Qj6A**
>
> Dear Reviewer Qj6A,
>
> Thank you for your recognition of our work.
>
>
> > **Q: On the implicit assumption $\|q_{t+1}^{(m)}\| = \|q_t^{(m)}\|$ in the angular bound (Appendix B).**
>
> We thank the reviewer for pointing out this issue. We have revised the proof as follows.
>
> In the revised appendix, we replace this step with a more general geometric
> inequality that does not assume equal norms.
>
> Specifically, for each channel $m$ we now let
> $$
> r_m := \min \\{ \|q_t^{(m)}\|,\|q_{t+1}^{(m)}\| \\} > 0,
> \qquad
> \varepsilon^{(m)} := \|q_{t+1}^{(m)} - q_t^{(m)}\|.
> $$
> In the 2D RoPE subspace, consider the angle
> $\Delta\phi^{(m)}:= \phi_{t+1,i}^{(m)} - \phi_{t,i}^{(m)}$ between
> $q_t^{(m)}$ and $q_{t+1}^{(m)}$.
> By a simple planar-geometry argument (projecting both vectors onto the circle
> of radius $r_m$ and using the law of cosines), one can show that
> $$
> 2 r_m \sin\Big(\tfrac{1}{2} |\Delta\phi^{(m)}|\Big)
> \le
> \varepsilon^{(m)}.
> $$
> Consequently,
> $$
> |\Delta\phi^{(m)}|
> \le
> 2 \arcsin\Big(\frac{\varepsilon^{(m)}}{2 r_m}\Big)
> \le
> \frac{\pi}{2}\ \frac{\varepsilon^{(m)}}{r_m}
> \le
> \frac{\pi}{2}\ \frac{\varepsilon}{r_m},
> $$
> where $\varepsilon = \|q_{t+1} - q_t\|$.
> This yields exactly the same type of bound used in Equation (10), but now
> without any equal-norm assumption.
>
> We have updated Appendix B accordingly:
>
> - the equation "$\varepsilon^{(m)} = 2 r_m \sin(\cdot)$" is replaced by the
>   inequality "$2 r_m \sin(\cdot) \le \varepsilon^{(m)}$" with
>   $r_m = \min\\{\|q_t^{(m)}\|,\|q_{t+1}^{(m)}\|\\}$;
> - the subsequent derivation is revised to use the above inequality.
>
> The statement and asymptotic conclusion of Theorem 5.1 (vertical stability)
> remain unchanged.
>
> ---
>
> We would also like to express our gratitude once again for your **careful reading** and **constructive suggestions**, which have significantly improved the quality of our work. **We greatly enjoy our exchange with you**.
>
> Best regards,
>
> Authors of Submission 22783

---

### Official Review · Reviewer_V8Hz · 2025-10-29

**Soundness:** 2
**Presentation:** 2
**Contribution:** 3
**Rating:** 4
**Confidence:** 3

**Summary:**

This paper analyzes sparse attention patterns from the perspective of query similarity. It derives theoretical relations between the similarity of consecutive query vectors and the corresponding changes in attention values. The paper further shows how certain query distributions lead to typical attention patterns. It shows potential applications of these insights in attention budget allocation and layer pruning.

**Strengths:**

1. The paper provides a new perspective for explaining existing attention patterns from the query point of view.

2. The paper demonstrates how the proposed query-level observations can inform the design of sparse attention and layer pruning, adding practical value to the theoretical analysis.

**Weaknesses:**

1. The claim of analyzing the *joint effect of input dynamics and positional encoding* seems overstated. While it would be valuable to disentangle and quantify their respective contributions, the paper instead merges them into the query with post-encoding. This makes the connection to the original input less clear than the abstract and introduction suggest.

2. Several assumptions used in the derivations are not carefully validated, which raises concerns about the reliability of the conclusions. For instance, the assumption of a dominant channel weight in Theorem 5.1 requires empirical support.

3. The empirical section lacks comparisons with more direct and recent baselines, such as DuoAttention [1], which explicitly distinguishes retrieval heads.

4. Some key concepts (e.g., continuity, predictability) are introduced without sufficient explanation in the introduction. Brief definitions would help prevent confusion.

[1] Xiao, Guangxuan, et al. "Duoattention: Efficient long-context llm inference with retrieval and streaming heads." arXiv preprint arXiv:2410.10819 (2024).

**Questions:**

1. Is the query similarity computed after applying RoPE?

2. Could you provide more justification for using attention patterns to guide layer-wise FFN pruning in Section 6.2? In particular, why does high query similarity (stability) imply that “the layer extracts less novel information”

---

> ### Author Response · Authors · 2025-11-24
> **Response to Reviewer V8Hz**
>
> Dear Reviewer V8Hz,
>
> Thank you for your **valuable review** and for your **recognition of our query-similarity-based perspective on attention patterns and its practical implications for downstream tasks**. Your positive evaluation is truly encouraging and means a great deal to us. Below, we provide detailed responses to your insightful and constructive comments. We have incorporated the corresponding discussions and experiments into the appendix and added the necessary references in the main text.
>
> Throughout the discussion phase, we genuinely look forward to your feedback and are fully committed to addressing any remaining concerns. If our responses have properly resolved your questions,  **we would be deeply grateful if you would consider raising your score** . If anything remains unclear or unconvincing,  **we would truly appreciate your further guidance** , and we stand ready to engage promptly and constructively.
> > **W1:** The claim of analyzing the *joint effect of input dynamics and positional encoding* seems overstated. While it would be valuable to disentangle and quantify their respective contributions, the paper instead merges them into the query with post-encoding. This makes the connection to the original input less clear than the abstract and introduction suggest.
>
> **A:** We thank the reviewer for this insightful comment. We understand the concern that treating the rotated query as a single entity might obscure the separate contributions of the original semantic input and the positional bias.
> - **Clarification on what is actually disentangled**: In our analysis we do not only look at the RoPE-rotated query as a whole. We first measure the temporal continuity of the **pre-RoPE queries**, and show that this continuity can separate predictable vs. unpredictable heads. We then study how RoPE, when applied on such (possibly special) query regimes, induces different geometric structures in the attention map and gives rise to different patterns. In this sense, the roles of “input-driven query dynamics” and “positional encoding” are explicitly separated in our framework.
> - **New decoupling on a sequential head (Appendix F.1)**:
>   - To clarify the roles of input dynamics and RoPE, we added an ablation on a representative sequential head. Using the same head and input, we compare: (1) high query similarity with RoPE (original model), (2) high query similarity **without RoPE** (rotation replaced by identity), and (3) RoPE with **mildly perturbed query dynamics**, where the average cosine similarity between queries is reduced from 0.99 to 0.97 by randomly resampling time indices.
>   - **What the three conditions show.**  (1) With high q-similarity and RoPE, the attention map forms an almost perfectly smooth diagonal stripe that shifts along the (+1,+1) direction. (2) Removing RoPE while keeping queries/keys fixed preserves only a coarse near-diagonal bias, with the diagonal broken into segments and superposed with vertical streaks. (3) Keeping RoPE but perturbing the queries yields a map that still has a diagonal tendency but is overlaid with many scattered, unpredictable activation spikes.
>   - **Implication for the ''joint effect'' claim.**  These results indicate that high q-similarity alone encourages local, near-diagonal attention but does not produce the clean, shift-invariant diagonal pattern, and RoPE alone (with perturbed queries) yields only a partially structured, more unpredictable pattern. Only when smooth query dynamics and RoPE are both present do we recover the stable sequential pattern of the full model, supporting our revised claim that we analyze how input dynamics and positional encoding together influence attention patterns.

---

> ### Author Response · Authors · 2025-11-24
> **Response to Reviewer V8Hz**
>
> > **W2:** Several assumptions used in the derivations are not carefully validated, which raises concerns about the reliability of the conclusions. For instance, the assumption of a dominant channel weight in Theorem 5.1 requires empirical support.
>
> **A:** We thank the reviewer for pointing out the need for empirical validation of assumptions.
>
> 1. **Alignment with Established Empirical Studies:** The assumption of a dominant weight ($w_m \approx ||k^{(m)}|| \cdot ||q^{(m)}||$) is grounded in the well-documented phenomenon of "Massive Activations" or "Outliers" in Transformer Key states. Prior works, such as KIVI [1] and KVQuant [2], have reported that specific feature channels in LLMs consistently exhibit magnitudes significantly larger than others.
> 2. **Empirical** **validation** **on re-access heads:** We include an new empirical study of the dominant-channel in **Appendix F.2**. For the re-access head, we visualize a RoPE-channel spectrum at the sink position (Figure 5). We observe a highly concentrated distribution: **one low-frequency channel already accounts for about 51% of the total mass**, while all remaining channels form a long, low-magnitude tail. This directly confirms that, for the re-access head under study, the attention logits are indeed dominated by a single low-frequency channel, as assumed in Theorem 5.1.
>
> [1] Liu et al., "KIVI: A Tuning-Free Asymmetric 2bit Quantization for KV Cache." ICML, 2024.
>
> [2] Hooper et al., "KVQuant: Towards 10 Million Context Length LLM Inference with KV Cache Quantization." NeurIPS, 2024.
>
> > **W3:** The empirical section lacks comparisons with more direct and recent baselines, such as DuoAttention [3], which explicitly distinguishes retrieval heads.
>
> **A:** We thank the reviewer for suggesting the comparison with the recent baseline, DuoAttention [3]. We have conducted additional experiments to compare our method against DuoAttention. The detailed results have been added to **Appendix H.**
>
> 1. **Superior Performance:** As shown in Table 1 below, **our method consistently achieves superior or comparable performance compared to DuoAttention** across different KV cache budgets. Notably, our method demonstrates better robustness in challenging tasks like HotpotQA.
> 2. **Training-Free Efficiency:** A critical advantage of our approach is its efficiency and simplicity.
>    - DuoAttention requires a computationally expensive pre-processing stage to identify retrieval heads (reported as 2,000 training steps on 8$\times$NVIDIA A100 GPUs).
>    - Ours is entirely **training-free**. The q-similarity metric is an intrinsic property calculated on-the-fly, incurring negligible overhead (approx. **0.2ms** and **9MB** memory per layer).
> 3. **Theoretical advantage:** Recent methods such as DuoAttention, KVTuner, and MInference effectively leverage the difference between retrieval-style and streaming-style attention to build strong compression. Our analysis provides two additional theoretical advantages:
>    - **Richer categorization of compressible attention patterns and potential for higher compression ratios.** Our q-similarity framework further refine several regular and predictable patterns from the times series persbective, and explicitly identifies these regular patterns as compressible. This expands the scope of heads that can be safely considered for compression beyond purely streaming behavior. As a result, our framework suggests the potential to achieve higher compression ratios at similar accuracy by also compressing these additional predictable heads.
>    - **Theoretical explanation of why predictable patterns emerge.** Our framework does not only categorize patterns, but also derives explicit conditions under which different predictable attention behaviors arise. This provides a theoretical account of *why* re-access, sequential, and seasonal heads appear in practice from a time-series perspective, and offers guidance for designing new attention mechanisms or hybrid architectures that explicitly exploit these predictable patterns.
>
> ***Table 1:** Performance comparison with DuoAttention on Llama-3.1-8B (LongBench).*
>
> | Budget | Method | MF-en | HotpotQA | QMSum | TriviaQA | Pre   | Lcc   | Avg.|
> | ------ | --- | ----- | -------- | ----- | -------- | ----- | ----- | ------ |
> |  | Full   | 53.78 | 55.04    | 25.33 | 91.25    | 99.50 | 63.36 | 64.71|
> | 512    | DuoAttention | 51.70 | 54.15    | 24.50 | 92.35    | 99.50 | 64.55 | 64.46|
> | 512    | Ours   | 51.63 | 54.53    | 24.57 | 92.35    | 99.50 | 64.56 | **64.52** |
> | 1024   | DuoAttention | 52.72 | 54.58    | 24.62 | 91.89    | 99.50 | 64.78 | 64.68|
> | 1024   | Ours   | 52.63 | 55.45    | 24.59 | 92.04    | 99.50 | 64.58 | **64.80** |

---

> ### Author Response · Authors · 2025-11-24
> **Response to Reviewer V8Hz**
>
> > **W4:** Some key concepts (e.g., continuity, predictability) are introduced without sufficient explanation in the introduction. Brief definitions would help prevent confusion.
>
> **A:** Thank you for the suggestion. We have revised the Introduction (specifically Paragraph 3) to embed concise definitions of these core concepts upon their first mention. The specific modifications are listed below:
>
> 1. Defining "Predictability" :
>    - Original: ...we systematically categorize attention patterns into predictable and unpredictable types...
>    - Revised: ...we systematically categorize attention patterns into predictable, **characterized by clear regularities**, and unpredictable types **that appear random**...
> 2. Defining "Continuity":
>
> - Original: We further explain their emergence by analyzing the temporal continuity of queries and keys in conjunction with RoPE.
> - Revised: We further explain their emergence by analyzing the temporal continuity of queries and keys—**defined as the self-similarity across consecutive time steps**—in conjunction with RoPE.
>
> > **Q1:** Is the query similarity computed after applying RoPE?
>
> **A:** We compute all query similarity metrics using **pre-RoPE queries**. This choice allows us to **disentangle** **the temporal continuity** of the semantic content from the effects introduced by **positional encoding**. In the revised version, we will add the following clarification at the beginning of Section 4 to address the reviewer’s concern: “Note that all query similarity metrics are computed on the pre-RoPE queries to strictly measure the temporal continuity of the latent semantics, independent of positional rotations.”
>
> > **Q2:** Could you provide more justification for using attention patterns to guide layer-wise FFN pruning in Section 6.2? In particular, why does high query similarity (stability) imply that “the layer extracts less novel information”
>
> **A:** We appreciate the opportunity to clarify the motivation behind using query similarity to guide pruning.
>
> 1. Scope Clarification: First, we wish to clarify that our method performs **structured layer-wise** **pruning** (removing the entire Transformer block, including both Self-Attention and FFN modules), rather than pruning FFNs exclusively.
> 2. **Redundancy in Layer** **Pruning**: The fundamental principle of layer pruning, as established in prior works like ShortGPT, relies on identifying "redundant layers." A layer is considered redundant if its input hidden states are highly similar to its output hidden states (measured by Block Influence), indicating that the layer contributes minimal transformation or "new information" to the hidden states.
> 3. Why High Q-Similarity Implies Redundancy: We propose Q-similarity as a critical indicator of layer importance based on the distinct roles of attention heads:
>
>    - Low Q-Similarity (Retrieval Heads): As demonstrated in our paper, low Q-similarity correlates with *retrieval patterns*. Recent studies (e.g., KVTuner [4], DuoAttention [3]) emphasize that **retrieval heads are essential for handling complex tasks** and long-context understanding. Therefore, layers containing these dynamic, low-similarity heads are critical and should be preserved.
>    - High Q-Similarity (Stable Patterns): Conversely, high Q-similarity points to stable, repetitive patterns like attention sinks. These are common and redundant across the model. Thus, **pruning layers with stable patterns is safer** and impacts performance much less than losing unique retrieval heads.
>
> Consequently, layers with high Q-similarity are less likely to possess the critical retrieval capabilities required for performance. By combining Block Influence with Q-similarity, we can more accurately identify and prune layers that are both transformationally inactive and functionally less critical.
>
> [3] Xiao, Guangxuan, et al. "Duoattention: Efficient long-context llm inference with retrieval and streaming heads." ICLR 2025.
>
> [4] Li, Xing, et al. "KVTuner: Sensitivity-Aware Layer-Wise Mixed-Precision KV Cache Quantization for Efficient and Nearly Lossless LLM Inference." ICML 2025.

---

> > ### Comment · Reviewer_V8Hz · 2025-11-27
> >
> > I appreciate the authors’ responses, including the additional empirical evidence, clarifications, and comparisons with prior work. My only remaining concern is that the notion of the ‘joint effect’ should be explicitly clarified at its first appearance in the introduction. This will ensure that readers clearly understand the scope and contributions of the work.
> > With these considerations, I am raising my score to 6. But I strongly encourage the authors to revise the introduction of the main paper during rebuttal, as well as future versions, to prevent potential misunderstandings.

---

> > > ### Author Response · Authors · 2025-12-02
> > > **Further Response to Reviewer V8Hz**
> > >
> > > Dear Reviewer V8Hz,
> > >
> > > Thank you for your recognition of our work and for the helpful suggestion about clarifying the notion of the "joint effect".
> > >
> > > In the revised manuscript, we have **added a new third paragraph** in the introduction to explicitly define what we mean by the joint effect of queries, keys, and RoPE at its first mention. Specifically, we now describe the attention logit as the dot product between the current query and the corresponding key after RoPE rotation, emphasize that keys and their RoPE rotations are fixed once stored in the cache while queries evolve over time, and explain that the visible attention patterns are governed by the interaction between query trajectories, key geometry, and dominant RoPE channels. These additions help readers see how it underpins our analytical framework developed in the paper and directly addresses the potential misunderstanding you pointed out.
> > >
> > > We would also like to express our gratitude once again for your **careful reading** and **constructive suggestions**, which have significantly improved the quality of our work.
> > >
> > > Best regards,
> > >
> > > Authors of Submission 22783

---

> ### Author Response · Authors · 2025-11-27
> **Gentle Reminder: Rebuttal and Updated Paper Available**
>
> Dear Reviewer V8Hz,
>
> This is a gentle reminder that we have submitted our rebuttal and updated the paper accordingly. If you have a moment, we would greatly appreciate it if you could take a look and share any further comments during the discussion phase. We are very happy to clarify any points or provide additional information as needed.
>
> Thank you very much for your time and for engaging with our submission.
>
> Best regards,
>
> Authors of Submission 22783

---

### Official Review · Reviewer_yc4e · 2025-10-31

**Soundness:** 2
**Presentation:** 3
**Contribution:** 2
**Rating:** 4
**Confidence:** 4

**Summary:**

This paper proposes a framework to explain diverse attention patterns in LLMs through temporal continuity analysis. The authors categorize patterns as predictable (re-access, sequential, seasonal) or unpredictable, attributing the distinction to query self-similarity. They provide mathematical analysis of how query/key continuity and RoPE jointly produce these patterns, and validate their framework through KV cache compression and (fewer) LLM pruning experiments. While the paper makes a reasonable attempt to unify attention pattern analysis, the contributions are incremental over existing work (particularly AttentionPredictor). The theoretical analysis, though rigorous, doesn't yield sufficiently novel insights—the sequential pattern analysis overstates its departure from prior work, and the seasonal pattern analysis lacks empirical grounding. The downstream experiments show only marginal improvements.

**Strengths:**

- The temporal continuity perspective provides a systematic way to understand previously fragmented observations about attention patterns. The decomposition view connecting query similarity to pattern stability is intuitive.
- Rigorous mathematical treatment: the theorems provide formal proofs for the emergence of different pattern types, with explicit bounds relating pattern stability to query/key properties and RoPE parameters.
- Novel insight on periodic sequential patterns: The analysis of diagonal spacing and experimental validation by manipulating dominant channel locations is particularly interesting.
- The evaluation on KV Cache compression is performed with different budgets.

**Weaknesses:**

- Limited novelty: the observation that query continuity drives attention stability was already made by AttentionPredictor (and the authors acknowledge that). While this paper provides mathematical formalization, the fundamental insight is not new.
- KV cache compression: the improvements over CAKE are marginal and seem to be within noise margins. Other state of art methods such as DuoAttention, Expected Attention could be a stronger baseline.
- LLM pruning is only compared against a single baseline. There are no other comparisons to other structured pruning methods. If the authors think this makes sense, could they explain why they considered only this one baseline.
- The hyperparameters  appear hand-tuned without ablation studies.

**Questions:**

- What is the computational overhead of computing q-similarity scores during inference?
- What is the precise impact of this KV cache compression method on memory footprint ? And on latency ?
- How are the hyper-parameters selected ?

---

> ### Author Response · Authors · 2025-11-24
> **Response to Reviewer yc4e**
>
> Dear Reviewer yc4e,
>
> Thank you for your **valuable review** and for your **recognition of our temporal-continuity framework, the theoretical development that formalizes this perspective, and the new insights**. Your positive evaluation is truly encouraging and means a great deal to us. Below, we provide detailed responses to your insightful and constructive comments. We have incorporated the corresponding discussions and experiments into the appendix and added the necessary references in the main text.
>
> Throughout the discussion phase, we genuinely look forward to your feedback and are fully committed to addressing any remaining concerns. If our responses have properly resolved your questions,  **we would be deeply grateful if you would consider raising your score** . If anything remains unclear or unconvincing,  **we would truly appreciate your further guidance** , and we stand ready to engage promptly and constructively.
>
>
> > **W1:** Limited novelty: the observation that query continuity drives attention stability was already made by AttentionPredictor (and the authors acknowledge that). While this paper provides mathematical formalization, the fundamental insight is not new.
>
> **A:** We appreciate the reviewer’s attention to the novelty of our work. Although AttentionPredictor introduces a useful analytical perspective, its analysis is limited to a subset of attention patterns and lacks systematic theoretical grounding, which restricts its practical applicability. In contrast, our work provides substantial advances in theoretical completeness, analytical depth, and practical value, as summarized below:
>
> 1. **Systematic and general framework with downstream validation.** We propose a unified framework based on query similarity to distinguish predictable vs. unpredictable patterns, together with explicit conditions for the emergence of three representative predictable patterns. The theoretical findings are model-agnostic, and can be instantiated to derive other training-free proxy metrics and guide additional applications such as training-free automatic precision allocation for weight quantization and critical-layer identification. The findings also theoretically validate the effectiveness of the trend of hybrid linear and full attention models. In the current paper, we instantiate this framework on two concrete downstream tasks—model pruning and KV cache compression—where the q-similarity–based metric consistently improves over strong baselines, an aspect not explored by AttentionPredictor.
> 2. **Refined analysis of the re-access pattern:** AttentionPredictor does not account for the effect of RoPE when analyzing the re-access pattern. We address and correct this limitation by providing complete derivations and emergence conditions under RoPE, making the conclusions consistent with real model behavior.
> 3. **New theory and visualization for the sequential pattern:** We derive a closed-form computation for sequential intervals and further validate it through visualization experiments.
> 4. **Novel theoretical explanation for the seasonal pattern:** We provide a new theoretical interpretation of the seasonal pattern and integrate it into a unified framework of attention behaviors.
>
> Finally, we note that the attention-analysis component of AttentionPredictor was released on about October 23, 2025, **contemporaneous with our submission**. Our work offers a more systematic, complete, and practically applicable theoretical analysis, further supported by downstream empirical validation.

---

> ### Author Response · Authors · 2025-11-24
> **Response to Reviewer yc4e**
>
> > **W2:** KV cache compression: the improvements over CAKE are marginal and seem to be within noise margins. Other state-of-the-art methods such as DuoAttention, Expected Attention could be a stronger baseline.
>
> **A:** We thank the reviewer for suggesting the comparison with the recent baseline, DuoAttention and Expected Attention. We have conducted additional experiments to compare our method against them. The detailed results have been added to Appendix H.
>
> 1. **DuoAttention:** We use the retrieval-head score provided by DuoAttention as a metric for allocating the layer-wise KV cache budget and compare it with our q-similarity–based allocation.
>    - **Superior Performance:** As shown in Table 1 below, our method consistently matches or outperforms DuoAttention across different budgets, and in particular shows better robustness on challenging long-context tasks such as HotpotQA.
>    - **Training-Free Efficiency:** A critical advantage of our approach is its efficiency and simplicity.
>      - DuoAttention requires a computationally expensive pre-processing stage to identify retrieval heads (reported as 2,000 training steps on 8$\times$NVIDIA A100 GPUs).
>      - Ours is entirely **training-free**. The q-similarity metric is an intrinsic property calculated on-the-fly, incurring negligible overhead (approx. **0.2ms** and **9MB** memory per layer).
>    - **Theoretical advantage:** Recent methods such as DuoAttention, KVTuner, and MInference effectively leverage the difference between retrieval-style and streaming-style attention to build strong compression. Our analysis provides two additional theoretical advantages:
>      - **Richer categorization of compressible attention patterns and potential for higher compression ratios.** Our q-similarity framework further refine several regular and predictable patterns from the times series persbective, and explicitly identifies these regular patterns as compressible. This expands the scope of heads that can be safely considered for compression beyond purely streaming behavior. As a result, our framework suggests the potential to achieve higher compression ratios at similar accuracy by also compressing these additional predictable heads.
>      - **Theoretical explanation of why predictable patterns emerge.** Our framework does not only categorize patterns, but also derives explicit conditions under which different predictable attention behaviors arise. This provides a theoretical account of *why* re-access, sequential, and seasonal heads appear in practice from a time-series perspective, and offers guidance for designing new attention mechanisms or hybrid architectures that explicitly exploit these predictable patterns.
>
>     ***Table 1:** Performance comparison with DuoAttention on Llama-3.1-8B (LongBench).*
>    | Budget | Method| MF-en | HotpotQA | QMSum | TriviaQA | Pre   | Lcc   | Avg.|
>    | ---- | ---- | ----- | -------- | ----- | -------- | ----- | ----- | ------ |
>    |  | **Full**   | 53.78 | 55.04| 25.33 | 91.25| 99.50 | 63.36 | 64.71|
>    | 512| **DuoAttention** | 51.70 | 54.15| 24.50 | 92.35| 99.50 | 64.55 | 64.46|
>    | 512| **Ours**   | 51.63 | 54.53| 24.57 | 92.35| 99.50 | 64.56 | **64.52** |
>    | 1024   | **DuoAttention** | 52.72 | 54.58| 24.62 | 91.89| 99.50 | 64.78 | 64.68|
>    | 1024   | **Ours** | 52.63 | 55.45| 24.59 | 92.04| 99.50 | 64.58 | **64.80** |
> 2. **Expected Attention:** We incorporate our q-similarity–based budget allocation strategy into the Expected Attention KV-cache compression framework. As shown in Table 2, adding the q-similarity budget allocation **consistently improves the performance of Expected Attention**. This demonstrates that our q-similarity metric can be integrated to enhance other compression frameworks, showing strong generality and compatibility. It further supports the universality of our proposed temporal perspective based on query continuity.
>
>   ***Table 2:** Performance comparison of Expected Attention (EA) with and without q-similarity budget allocation on Llama-3.1-8B (LongBench).*
>   | Budget | Method | MF-en | HotpotQA | QMSum | TriviaQA | Pre   | Lcc   | Avg.  |
>   | ------ | --- | ----- | -------- | ----- | -------- | ----- | ----- | ----- |
>   |  | Full   | 53.78 | 55.04| 25.33 | 91.25| 99.50 | 63.36 | 64.71 |
>   | 512| DuoAttention | 51.70 | 54.15| 24.50 | 92.35| 99.50 | 64.55 | 64.46 |
>   | 512| Ours   | 51.63 | 54.53| 24.57 | 92.35| 99.50 | 64.56 | 64.52 |
>   | 1024   | DuoAttention | 52.72 | 54.58| 24.62 | 91.89| 99.50 | 64.78 | 64.68 |
>   | 1024   | Ours   | 52.63 | 55.45| 24.59 | 92.04| 99.50 | 64.58 | 64.80 |

---

> ### Author Response · Authors · 2025-11-24
> **Response to Reviewer yc4e**
>
> > **W3:** LLM pruning is only compared against a single baseline. There are no other comparisons to other structured pruning methods. If the authors think this makes sense, could they explain why they considered only this one baseline.
>
> **A:** Regarding the choice of baselines, we agree that a more comprehensive comparison is necessary. We have therefore added experiments with several mainstream LLM pruning methods. These additional baselines follow the definitions and reported results mentioned in ShortGPT [1], including:
>
> - **LLMPruner** [2]: a gradient-based structured pruning method that removes non-critical coupled structures to preserve model functionality. Although the original method includes post-training, we follow the ShortGPT setup and exclude post-training for fair comparison.
> - **SliceGPT** [3]: a post-training pruning scheme that compresses the model by applying PCA to hidden states in each layer and reducing the embedding dimension.
> - **LaCo** [4]: a structured pruning approach based on layer merging, which gradually merges similar layers from deep to shallow while using a threshold to avoid excessive merging.
>
> We reproduced and summarized the results of these methods on Llama2-7B and conducted a systematic comparison with our approach. The results in **Table 3** show that our method achieves **better performance under higher pruning ratio**, demonstrating its effectiveness in LLM pruning scenarios.
>
> ***Table 3**: Comparison between additional mainstream LLM Pruning methods and our method. Experiments are conducted on Llama2-7B.*
>
> | Method  | Pruning Ratio| PIQA| Hellaswag | WSC | BoolQ| Race-H| Avg.|
> | - | ------- | ------ | ------ | ------ | ------ | ------ | ------ |
> | **LLMPruner** | 27.00%| 71.22| 56.46| 36.54| 55.20| 22.56| 48.40|
> | **SliceGPT**  | 26.40%| 66.21| 50.27| 36.54| 38.32| 21.07| 42.48|
> | **LaCo**| 27.10%| 69.80| 55.69| 40.38| 64.07| 22.61| 50.51|
> | **ShortGPT**  | 27.10%| 66.43| 53.02| 52.46| **74.71** | 32.25| 55.77|
> | **Ours**| **28.10%** | **66.76** | **55.97** | **68.13** | 62.17| **33.88** | **57.38** |
>
> [1] ShortGPT: Layers in Large Language Models are More Redundant Than You Expect. ACL 2025.
>
> [2] LLM-Pruner: On the Structural Pruning of Large Language Models. NeurIPS 2023.
>
> [3] SliceGPT: Compress Large Language Models by Deleting Rows and Columns. ICLR 2024.
>
> [4] LaCo: Large Language Model Pruning via Layer Collapse. EMNLP 2024.

---

> ### Author Response · Authors · 2025-11-24
> **Response to Reviewer yc4e**
>
> > **W4 & Q3:** Hyper-parameters appear hand-tuned without ablation studies. How are the hyper-parameters selected?
>
> **A:** Thank you for the suggestion. We add hyper-parameter sensitivity studies and clarify our hyper-parameter selection process. Overall, our method is not sensitive to the major hyper-parameters, and the results consistently show that larger weighting coefficients lead to better performance, further confirming the effectiveness of our approach.
>
> - **For** **KV** **cache compression task**, We conduct a hyper-parameter sensitivity study of $\alpha$. The results in **Table 5** show that the method remains stable across different $\alpha$ values, indicating **low sensitivity**. Moreover, larger $\alpha$ yields **consistently better performance**, demonstrating the usefulness of our proposed q-similarity. **During actual hyper-parameter selection**, for each model we evaluate $\alpha$ ∈ {0.1, 1, ∞} on a small validation set and select the best $\alpha$ as a unified hyper-parameter for this model, and use it to evaluate on all datasets and KV budgets.
>
>   ***Table 5**: Hyper-parameter ablation result of $\alpha$ using Llama-3.1-8B model under 1024 budget. The results are tested on two subsets (MF-en and HotpotQA) of the Longbench dataset.*
>
>   | $\alpha$    | MF-en| HotpotQA  | Avg.|
>   | ---- | ------ | ------ | ------ |
>   | **0**   | 52.16| 55.43| 53.80|
>   | **0.1** | 52.19| 55.43| 53.81|
>   | **0.2** | 52.17| 55.39| 53.78|
>   | **0.5** | 52.17| 55.39| 53.78|
>   | **0.8** | 52.22| 55.43| 53.83|
>   | **1**   | 52.14| 55.43| 53.79|
>   | **1.5** | 52.19| 55.42| 53.80|
>   | **2**   | 52.27| 55.42| 53.84|
>   | **5**   | 52.11| **55.56** | 53.84|
>   | **10**  | 52.24| 55.54| 53.89|
>   | **100** | **52.41** | 55.44| **53.93** |
> - **For layer** **pruning** **task**, we performed a similar hyper-parameter **$\beta$** sensitivity study. As shown in **Table 6**, the performance varies only slightly across different $\beta$ values, suggesting that the method is also insensitive to this parameter. **During actual hyper-parameter selection**, for each model we test $\beta$ ∈ {0.1, 0.3, 0.5} on a small validation set and use the best $\beta$ as a unified hyper-parameter for this model, and use it to evaluate on all datasets and pruning ratios in the full experiments.
>
>   * **Table 6:** Hyper-parameter ablation result of $\beta$ using Llama-2-7B model under 34% pruning ratio. The consistent results under some hyperparameters are because the corresponding pruning layers are the same.*
>
>   | $\beta$| PIQA| Hellaswag | Winogrande| Arc_Easy  | Average   |
>   | ---- | ------ | ------ | ------ | ------ | ------ |
>   | **0**   | 60.83| 42.11| 60.38| **44.15** | 51.87|
>   | **0.1** | 60.83| 42.11| **60.38** | **44.15** | 51.87|
>   | **0.2** | 60.45| **48.53** | 62.43| 42.55| **53.49** |
>   | **0.3** | 60.45| **48.53** | 62.43| 42.55| **53.49** |
>   | **0.4** | 60.45| **48.53** | 62.43| 42.55| **53.49** |
>   | **0.5** | **61.10** | 46.47| 57.77| 39.69| 51.26|

---

> ### Author Response · Authors · 2025-11-24
> **Response to Reviewer yc4e**
>
> > **Q1:** What is the computational overhead of computing q-similarity scores during inference?
>
> **A:** We conduct a systematic runtime and memory analysis under different input lengths and compare our method with CAKE. On Llama-3.1-8B with window size 32, the per-layer latency of computing q-similarities is below **0.2 ms** across context lengths up to 32K tokens, and the additional memory consumption is about **8 MB**, which is **practically negligible** (**Table 7**). The key reason for this low and stable overhead is that :
>
> - Our method only computes **query** **similarities within a fixed window**, so its complexity depends on the query dimension and the window size, but is essentially **independent of the total sequence length**. This also makes it naturally scalable to larger models or MoE models: as long as the query dimension does not change, the cost of our q-similarity computation remains the same.
> - In contrast, sota methods like CAKE computes its eviction metrics from **historical attention scores**, maintaining running statistics (mean and variance) for every cached token in each layer/head. This design requires storing and updating per-token statistics at every decoding step, so both its runtime and memory **overhead grow linearly with the context length**.
>
> Empirically, in long-sequence scenarios such as 32K tokens, our method reduces the per-layer latency overhead by up to **78%** and the additional memory consumption by up to **98%** compared to CAKE (see **Table 7**), highlighting the efficiency advantage of query-based eviction over attention-based approaches.
>
> ***Table 7:** Per-layer computational overhead comparison between our method and CAKE under different input lengths. All delays are measured in ms, and all memory consumption is measured in MB.*
>
> | Length  | CAKE Latency | CAKE Memory | Q-sim Latency | Q-sim Memory | Latency Improv. | Memory Improv. |
> | ---- | --- | -- | - | --- | -- | - |
> | **4K**  | 0.198  | 72.12 | 0.195   | 8.69   | 2%  | 88%|
> | **8K**  | 0.308  | 136.12| 0.197   | 8.69   | 36% | 94%|
> | **16K** | 0.506  | 264.12| 0.199   | 8.69   | 61% | 97%|
> | **32K** | 0.874  | 520.12| 0.196   | 8.69   | **78%**   | **98%**  |
>
> > **Q2:** What is the precise impact of this KV cache compression method on memory footprint? And on latency?
>
> **A:** To quantify the efficiency gains, we measured the per-token inference latency and KV cache memory footprint across varying input lengths. The results are summarized in **Table 8**.  At 32K context length with a 1K budget, we achieve a **2.3$\times$ speedup** and **32$\times$ memory reduction**.
>
> ***Table 8**:  Per-token latency and KV Cache memory footprint of Full Cache and our KV Cache compression. The KV budget of ours is 1K. All latencies are measured in ms, and all memory consumption is measured in MB.*
>
> | Context Length | Full Cache Latency | Ours Latency | Speed up | Full Cache Memory | Our Memory |
> | - | - | --- | - | ---- | - |
> | **4K**   | 35| 34| 1  | 512   | 128  |
> | **8K**   | 42| 34| 1.2| 1024  | 128  |
> | **16K**  | 51| 34| 1.5| 2048  | 128  |
> | **32K**  | 79| 34| 2.3| 4096  | 128  |

---

> ### Author Response · Authors · 2025-11-27
> **Gentle Reminder: Rebuttal and Updated Paper Available**
>
> Dear Reviewer yc4e,
>
> This is a gentle reminder that we have submitted our rebuttal and updated the paper accordingly. If you have a moment, we would greatly appreciate it if you could take a look and share any further comments during the discussion phase. We are very happy to clarify any points or provide additional information as needed.
>
> Thank you very much for your time and for engaging with our submission.
>
> Best regards,
>
> Authors of Submission 22783

---

### Official Review · Reviewer_UTJo · 2025-11-03

**Soundness:** 2
**Presentation:** 2
**Contribution:** 2
**Rating:** 4
**Confidence:** 3

**Summary:**

This paper proposes a framework for analysing attention patterns in Transformer-based models; author identify three types of patterns (namely re-access, sequential, and seasonal) from a subset of the heads (referred to as "unpredictable") based on query self-similarity and positional embeddings. Based on the proposed framework, authors then propose a method for efficiently allocate KV cache budgets and structured layer pruning, improving over baselines like CAKE and ShortGPT.

**Strengths:**

- The analysis in e.g. Proposition 4.1 that links attention stability to query self-similarity, and how query drift induces changes in the logit changes, is novel and interesting; likewise for Th. 5.2 and 5.3, which provide conditions under which sequential/periodic diagonals appear
- Using q-similarity in CAKE and ShortGPT yields sigificant improvements in several settings

**Weaknesses:**

- Improvements in CAKE (Tab. 1) seem very marginal, are they statistically significant? Averages are not clearly reported
- Computing q-similarities for every layer/head seems computationally expensive, but runtimes/costs are not discussed in-depth

**Questions:**

- Can you please expand more on the runtime/costs of computing q-similarities and what kind of overheads they add to methods like CAKE?
- Are there cases where q-similarity fails to misidentify retrieval heads? What does it happen in those scenarios?

---

> ### Author Response · Authors · 2025-11-24
> **Response to Reviewer UTJo**
>
> Dear Reviewer UTJo,
>
> Thank you for your **valuable review** and for your **recognition of the novelty of our theoretical analysis and the empirical effectiveness of our method over existing baselines**. Your positive evaluation is truly encouraging and means a great deal to us. Below, we provide detailed responses to your insightful and constructive comments. We have incorporated the corresponding discussions and experiments into the appendix and added the necessary references in the main text.
>
> Throughout the discussion phase, we genuinely look forward to your feedback and are fully committed to addressing any remaining concerns. If our responses have properly resolved your questions,  **we would be deeply grateful if you would consider raising your score** . If anything remains unclear or unconvincing,  **we would truly appreciate your further guidance** , and we stand ready to engage promptly and constructively.
>
> > **W1:** Improvements in CAKE (Tab. 1) seem very marginal, are they statistically significant? Averages are not clearly reported.
>
> **A:** We thank the reviewer for the valuable comment. Regarding the concern about the magnitude of improvements, we **conduct 3 independent runs** with our method and report the corresponding means and variances in Table 1. As shown, our method exhibits stable and consistent improvements over the baseline, with small variances across runs, indicating strong statistical robustness.
>
> Moreover, since **our accuracy is already very close to that of the full cache** (i.e., no compression), **achieving additional gains in this context is challenging**. Even so, under the 512-budget setting, our method still reduces the accuracy drop by **53%** compared to the SOTA method CAKE, demonstrating that our approach delivers a meaningful and effective improvement.
>
> ***Table 1**: Performance comparison between CAKE and our method under different cache budgets. Results of our method are reported as mean ± variance over 3 independent runs.*
>
> | Budget | Method | MF-en       | HotpotQA    | QMSum       | TriviaQA    | Pre         | Lcc         | Avg.        |
> | ------ | ------ | ----------- | ----------- | ----------- | ----------- | ----------- | ----------- | ----------- |
> |        | Full   | 53.78       | 55.04       | 25.33       | 91.25       | 99.5        | 63.36       | 64.71       |
> | 512    | CAKE   | 51.65       | 54.37       | 24.94       | 91.54       | 99.5        | 62.3        | 64.05       |
> | 512    | Ours   | 51.59±0.00 | 54.46±0.07 | 24.51±0.00 | 91.98±0.10 | 99.50±0.00 | 64.36±0.04 | 64.40±0.01 |
> | 1024   | CAKE   | 52.16       | 55.43       | 24.94       | 91.94       | 99.5        | 64.95       | 64.82       |
> | 1024   | Ours   | 52.18±0.00 | 55.41±0.02 | 24.97±0.01 | 92.21±0.05 | 99.50±0.00 | 65.02±0.00 | 64.88±0.00 |
> | 2048   | CAKE   | 53.57       | 55.5        | 24.67       | 91.48       | 99.5        | 63.23       | 64.05       |
> | 2048   | Ours   | 53.27±0.04 | 55.49±0.00 | 24.80±0.02 | 91.54±0.01 | 99.50±0.00 | 65.02±0.01 | 64.94±0.00 |

---

> ### Author Response · Authors · 2025-11-24
> **Response to Reviewer UTJo**
>
> > **W2 & Q1:** Computing q-similarities for every layer/head seems computationally expensive, but runtimes/costs are not discussed in-depth. Can you please expand more on the runtime/costs of computing q-similarities and what kind of overheads they add to methods like CAKE?
>
> **A:** We conduct a systematic runtime and memory analysis under different input lengths and compare our method with CAKE. On Llama-3.1-8B with window size 32, the per-layer latency of computing q-similarities is below **0.2 ms** across context lengths up to 32K tokens, and the additional memory consumption is about **8 MB**, which is **practically negligible** (**Table 2**). The key reason for this low and stable overhead is that :
>
> - Our method only computes **query** **similarities within a fixed window**, so its complexity depends on the query dimension and the window size, but is essentially **independent of the total sequence length**. This also makes it naturally scalable to larger models or MoE models: as long as the query dimension does not change, the cost of our q-similarity computation remains the same.
> - In contrast, sota methods like CAKE computes its eviction metrics from **historical attention scores**, maintaining running statistics (mean and variance) for every cached token in each layer/head. This design requires storing and updating per-token statistics at every decoding step, so both its runtime and memory **overhead grow linearly with the context length**.
>
> Empirically, in long-sequence scenarios such as 32K tokens, our method reduces the per-layer latency overhead by up to **78%** and the additional memory consumption by up to **98%** compared to CAKE (see **Table 2**), highlighting the efficiency advantage of query-based eviction over attention-based approaches.
>
> ***Table 2:** Per-layer computational overhead comparison between our method and CAKE under different input lengths using Llama-3.1-8B model. The window size is fixed at 32. All delays are measured in ms, and all memory consumption is measured in MB.*
>
> | Length | CAKE Latency | CAKE Memory | Q-sim Latency | Q-sim Memory | Latency Improv. | Memory Improv. |
> | ------ | ------------ | ----------- | ------------- | ------------ | --------------- | -------------- |
> | 4K     | 0.198        | 72.12       | 0.195         | 8.69         | 2%              | 88%            |
> | 8K     | 0.308        | 136.12      | 0.197         | 8.69         | 36%             | 94%            |
> | 16K    | 0.506        | 264.12      | 0.199         | 8.69         | 61%             | 97%            |
> | 32K    | 0.874        | 520.12      | 0.196         | 8.69         | 78%             | 98%            |
>
> > **Q2:** Are there cases where q-similarity fails to misidentify retrieval heads? What does it happen in those scenarios?
>
> **A:** Yes, q-similarity can occasionally disagree with the retrieval behavior, but in our experiments these cases are very rare and have limited impact on compression quality. Concretely, we say a failure happens when a head that we categorize as retrieval-like is not flagged as such by our q-similarity score. On Llama-3.1-8B and LongBench dataset, q-similarity correctly identifies the vast majority of retrieval-like heads, and only roughly 7% of them are missed.
>
> Most mismatches fall into two simple corner cases:  (i) **High q-similarity but retrieval behavior**, where queries are stable but the relevant keys appear sparsely in the context, leading to occasional jumps. (ii) **Low q-similarity but stable behavior**, where strong attention sinks dominate even though queries drift over time.
>
> The practical consequences of these failures are limited. In case (i), we may allocate a slightly smaller budget to a retrieval-like head, but our method is always combined with base policies such as H2O or SnapKV that retain heavy-hitter tokens, so the actually retrieved tokens are typically preserved. In case (ii), our metric simply allocates more budget than necessary, which is a safe failure that mildly reduces compression efficiency but does not hurt accuracy. Overall, the small empirical failure ratio and the strong end-to-end results together indicate that q-similarity is a reliable proxy for identifying retrieval heads in practice.

---

> ### Author Response · Authors · 2025-11-27
> **Gentle Reminder: Rebuttal and Updated Paper Available**
>
> Dear Reviewer UTJo,
>
> This is a gentle reminder that we have submitted our rebuttal and updated the paper accordingly. If you have a moment, we would greatly appreciate it if you could take a look and share any further comments during the discussion phase. We are very happy to clarify any points or provide additional information as needed.
>
> Thank you very much for your time and for engaging with our submission.
>
> Best regards,
>
> Authors of Submission 22783

---

### Comment · Area_Chair_kuVu · 2025-11-27

Dear reviewers,

A reminder that the discussion phase will end in a few days (**December 2**). Engaging with the author's rebuttal is essential to address all potential concerns before our final discussion stage.

Thanks,
The AC

---

### Meta-Review · Area_Chair_CuTF · 2025-12-26

**Summary:**

This paper offers a coherent temporal perspective on attention patterns and develops it with a level of mathematical care that improved substantially during rebuttal. The authors clarified the scope of their assumptions, corrected several nontrivial proof issues, and added stronger baselines, ablations, and overhead analyses. The empirical gains — primarily in KV-cache compression and layer pruning — are numerically modest but consistent, training-free, and achieved with very low overhead, which makes them practically relevant. Beyond the numbers, there is also genuine intellectual value in trying to unify and formalize where the now-familiar attention patterns actually come from: the paper provides a structured lens that connects observed behaviors to continuity assumptions and RoPE geometry. The conceptual novelty is incremental relative to earlier work such as AttentionPredictor and retrieval/streaming-style analyses, but it goes further in organizing and formalizing these ideas and tying them to concrete mechanisms. Taken together, this seems like a reasonable accept as a careful explanatory-and-application contribution with a thoughtful attempt at conceptual unification.

**Reviewer Concerns:**

Addressed:
The rebuttal clarified definitions and scope, corrected key issues in several proofs, and added stronger baselines, ablations, and overhead analyses. Concerns about runtime, q-similarity robustness, empirical validation of assumptions (e.g., dominant channels, continuity), and pruning justification were largely resolved, which led two reviewers to raise their scores.

Still outstanding:
The remaining hesitation is mostly conceptual rather than technical: the empirical gains, while consistent, are modest, and the degree of novelty relative to prior work (e.g., AttentionPredictor and retrieval/streaming analyses) remains debatable. These issues are mitigated but not fully eliminated by the rebuttal.

**Reviewer Scores:**

UTJo (4) – Technical clarifications and overhead analysis help, but small gains remain → likely unchanged.
yc4e (4) – Added baselines/ablations address many points, but novelty concern persists → maybe +1, more likely unchanged.
V8Hz (4 → 6) – Concerns about “joint effect” and assumptions were addressed → raised to ~6.
Qj6A (4 → 6) – Proof corrections and added analyses resolved main issues → raised to ~6.

---

### Decision · Program_Chairs · 2026-01-26

Accept (Poster)